# A conserved major facilitator superfamily member orchestrates a subset of O-glycosylation to aid macrophage tissue invasion

Katarina Valoskova[1†], Julia Biebl[1†], Marko Roblek[1], Shamsi Emtenani[1], Attila Gyoergy[1], Michaela Misova[1], Aparna Ratheesh[1,2], Patricia Reis-Rodrigues[1], Kateryna Shkarina[1‡], Ida Signe Bohse Larsen[3], Sergey Y Vakhrushev[3], Henrik Clausen[3], Daria E Siekhaus[1]*

[1]Institute of Science and Technology Austria, Klosterneuburg, Austria; [2]Centre for Mechanochemical Cell Biology and Division of Biomedical Sciences, Warwick Medical School, University of Warwick, Coventry, United Kingdom; [3]Copenhagen Center for Glycomics, Department of Cellular and Molecular Medicine, Faculty of Health Sciences, University of Copenhagen, Copenhagen, Denmark

**Abstract** Aberrant display of the truncated core1 O-glycan T-antigen is a common feature of human cancer cells that correlates with metastasis. Here we show that T-antigen in *Drosophila melanogaster* macrophages is involved in their developmentally programmed tissue invasion. Higher macrophage T-antigen levels require an atypical major facilitator superfamily (MFS) member that we named Minerva which enables macrophage dissemination and invasion. We characterize for the first time the T and Tn glycoform O-glycoproteome of the *Drosophila melanogaster* embryo, and determine that Minerva increases the presence of T-antigen on proteins in pathways previously linked to cancer, most strongly on the sulfhydryl oxidase Qsox1 which we show is required for macrophage tissue entry. Minerva's vertebrate ortholog, MFSD1, rescues the *minerva* mutant's migration and T-antigen glycosylation defects. We thus identify a key conserved regulator that orchestrates O-glycosylation on a protein subset to activate a program governing migration steps important for both development and cancer metastasis.
DOI: https://doi.org/10.7554/eLife.41801.001

*For correspondence:
daria.siekhaus@ist.ac.at

†These authors contributed equally to this work

Present address: ‡Department of Biochemistry, The University of Lausanne, Epalinges, Switzerland

Competing interests: The authors declare that no competing interests exist.

## Introduction

The set of proteins expressed by a cell defines much of its potential capacities. However, a diverse set of modifications can occur after the protein is produced to alter its function and thus determine the cell's final behavior. One of the most frequent and variable of such alterations is glycosylation, in which sugars are added onto the oxygen (O) of a serine or threonine or onto the nitrogen (N) of an asparagine (*Kornfeld and Kornfeld, 1985*; *Marshall, 1972*; *Ohtsubo and Marth, 2006*). O-linked addition can occur on cytoplasmic and nuclear proteins in eukaryotes (*Comer and Hart, 2000*; *Hart et al., 2011*), but the most extensive N- and O- linked glycosylation occurs during the transit of a protein through the secretory pathway. A series of sugar molecules are added starting in the endoplasmic reticulum (ER) or cis-Golgi and continuing to be incorporated and removed until passage through the trans Golgi network is complete (*Aebi, 2013*; *Stanley et al., 2009*). N-linked glycosylation is initiated in the ER at consensus NxS/T X≠P site, whereas the most common GalNAc-type O-linked glycosylation is initiated in the early Golgi and glycosites display no clear sequence motifs, apart from a prevalence of neighboring prolines (*Bennett et al., 2012*; *Thanka Christlet and*

**eLife digest** Proteins, the workhorses of the body, participate in virtually every single process in a cell. Different types of molecules, such as sugars, can be added onto a protein to change its role or location, but this process may also play a role in cancer. Indeed, tumor cells that contain certain sugar modifications are more likely to be able to spread through the body. For example, a specific combination of sugars called T antigen is rarely present in healthy adult cells; yet, it is commonly found in cancer cells that leave the tumor where they were born and invade another tissue to form a new tumor. However, it is not clear whether T antigen actively helps this process inside the body, or is simply present during it.

To answer this question, Valoskova, Biebl et al. used genetic and biochemistry tools to study developing fruit fly embryos, where certain immune cells carry T antigen on their proteins. Like invading cancer cells, these immune cells can get inside tissues during development. The experiments revealed that a protein called Minerva helps attach T antigen onto proteins. When embryos were engineered to contain less Minerva, the amount of T antigen in the immune cells dropped, and the cells could not easily make their way into tissues anymore. When the mouse version of Minerva was then added to the embryos, the immune cells of the fruit flies had higher T antigen levels on their proteins and could invade tissues again.

Some of the proteins targeted by Minerva were known to be involved in cancer, but not all of them. Future experiments will investigate which role the human version of Minerva plays in cancer cells that get inside new tissues, and if it could help us predict whether a cancer is likely to spread.
DOI: https://doi.org/10.7554/eLife.41801.002

*Veluraja, 2001*). Glycosylation can affect protein folding, stability and localization as well as serve specific roles in fine-tuning protein processing and functions such as protein adhesion and signaling (*Goth et al., 2018*; *Varki, 2017*). The basic process by which such glycosylation occurs has been well studied. However our understanding of how specific glycan structures participate in modulating particular cellular functions is still at its beginning.

The need to understand the regulation of O-glycosylation is particularly relevant for cancer (*Fu et al., 2016*; *Häuselmann and Borsig, 2014*). The truncated O-glycans called T and Tn antigen are not normally found on most mature human cells (*Cao et al., 1996*) but up to 95% of cells from many cancer types display these at high levels (*Boland et al., 1982*; *Cao et al., 1996*; *Howard and Taylor, 1980*; *Limas and Lange, 1986*; *Orntoft et al., 1985*; *Springer, 1984*; *Springer et al., 1975*). The T O-glycan structure (Galβ1-3GalNAcα1-O-Ser/Thr) is synthesized by the large family of polypeptide GalNAc-transferases (GalNAc-Ts) that initiate protein O-glycosylation by adding GalNAc to form Tn antigen and the core1 synthase C1GalT1 that adds Gal to the initial GalNAc residues (*Tian and Ten Hagen, 2009*) to form T antigen (*Figure 1A*). The human C1GalT1 synthase requires a dedicated chaperone, COSMC, for folding and ER exit (*Ju and Cummings, 2005*). In adult humans these O-glycans are normally capped by sialic acids and/or elongated and branched into complex structures (*Tarp and Clausen, 2008*). However, in cancer this elongation and branching is reduced or absent and the appearance of these truncated T and Tn O-glycans correlates positively with cancer aggressiveness and negatively with long-term prognoses for many cancers in patients (*Baldus et al., 2000*; *Carrasco et al., 2013*; *Ferguson et al., 2014*; *MacLean and Longenecker, 1991*; *Schindlbeck et al., 2005*; *Springer, 1997*; *Springer, 1989*; *Summers et al., 1983*; *Yu et al., 2007*). The molecular basis for the enhanced appearance of T antigen in cancers is not clear (*Chia et al., 2016*), although higher Golgi pH in cancer cells correlates with increases in T antigen (*Kellokumpu et al., 2002*). Interestingly, T antigen is also observed as a transient fetal modification (*Barr et al., 1989*) and cancer cells frequently recapitulate processes that happened earlier in development (*Cofre and Abdelhay, 2017*; *Pierce, 1974*). Identifying new mechanisms that regulate T antigen modifications developmentally has the potential to lead to insights into cancer biology.

*Drosophila* as a classic genetic model system is an excellent organism in which to investigate these questions. *Drosophila* displays T antigen as the predominant form of GalNAc-, or mucin-type, O-glycosylation in the embryo with 18% of the T glycans being further elaborated, predominantly by the addition of GlcA (*Aoki et al., 2008*). As in vertebrates, the GalNAc-T isoenzymes directing the

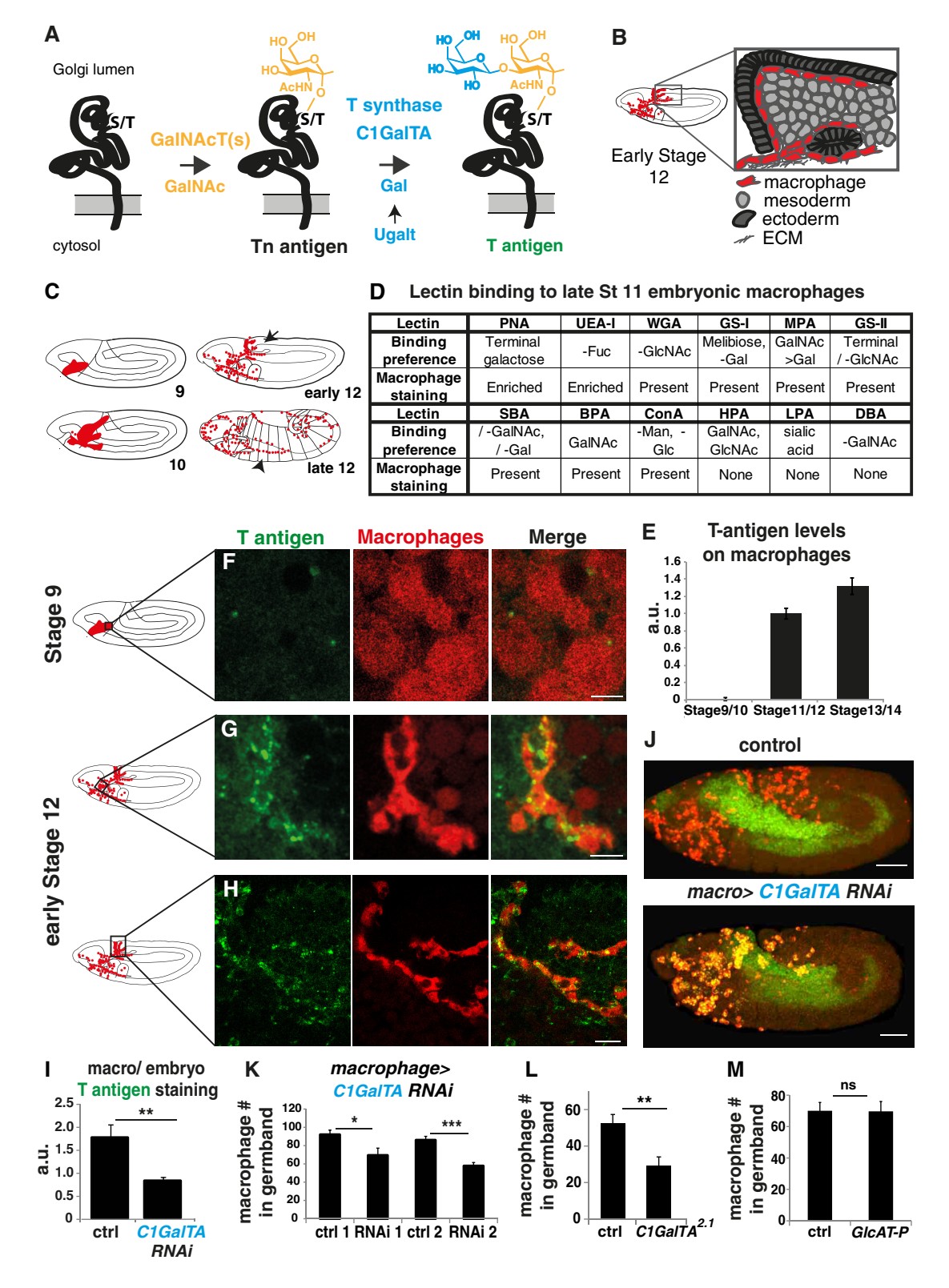

**Figure 1.** T antigen is enriched on *Drosophila* macrophages prior to and during their invasion of the extended germband. (**A**) Schematic of T antigen modification of serine (**S**) and threonine (**T**) on proteins within the Golgi lumen, through successive addition of GalNAc (yellow) by GalNAcTs and Gal (blue) by C1GalTs. Ugalt transports Gal into the Golgi. Glycosylation is shown at a much larger scale than the protein. (**B**) Schematic of an early Stage 12 embryo and a magnification of macrophages (red) entering between the germband ectoderm (dark grey), and mesoderm (light grey). (**C**) Schematic

*Figure 1 continued on next page*

*Figure 1 continued*

showing macrophages (red) disseminating from the head mesoderm in Stage 9. By Stage 10, they migrate towards the extended germband, the dorsal vessel and along the ventral nerve cord (vnc). At late Stage 11 germband invasion (arrow) begins and continues during germband retraction. Arrowhead highlights migration along the vnc in late Stage 12. (D) Table summarizing a screen of glycosylation-binding lectins for staining on macrophages invading the germband in late Stage 11 embryos. The listed binding preferences are abbreviated summaries of the specificities defined with mammalian glycans or simple saccharides which may have only incomplete relevance to insect glycomes. Enrichment was seen for PNA which recognizes T antigen and UEA-I which can recognize fucose. (E) Quantification of T antigen fluorescence intensities on wild type embryos shows upregulation on macrophages between Stage 9/10 and Stage 11/12. Arbitrary units (au) normalized to one for Stage 11. p<0.0001. (F–H) Confocal images of fixed lateral wild type embryos from (F) Stage 9 and (G–H) early Stage 12 with T antigen visualized by antibody staining (green) and macrophages by *srpHemo-3xmCherry* expression (red). Schematics at left with black boxes showing the imaged regions. (I) Quantification of control shows T antigen enrichment on macrophages when normalized to whole embryo. RNAi in macrophages against C1GalTA by *srpHemo(macrophage) >C1 GalTA RNAi vdrc2826* significantly decreases this T antigen staining (n = 8 embryos, p=0.011). (J) Representative confocal images of Stage 12 embryos from control and the aforementioned C1GalTA RNAi. Macrophages marked with cytoplasmic GFP (red) and nuclear RFP (green). (K,L) Quantification of macrophages in the germband in Stage 12 embryos for (K) control and two independent RNAis against C1GalTA (*vdrc110406* or *vdrc2826*) expressed in macrophages by the *srpHemo-Gal4* driver (n = 21–31 embryos, p<0.0001 and 0.017) or (L) in control and the *C1GalTA[2.1]* excision mutant (n = 23–24, p=0.0006). Macrophages labeled with *srpHemo-H2A::3xmCherry*. The RNAis and the mutant significantly decreased the macrophage number, arguing that T antigen is required in macrophages for germband entry. (M) Quantification of germband macrophages in early Stage 12 embryos in control and *GlcAT-P^{MI05251}* embryos shows no defect in macrophage invasion in the mutant (n = 17–20, p=0.962). (E) analyzed by Kruskal-Wallis test I, K-M analyzed by Student's t-test. ns = p > 0.05, *p<0.05; **p<0.01; ***p<0.001. Scale bars represent 10 μm in F–H, and 50 μm in J. See also *Figure 1—figure supplement 1*.

DOI: https://doi.org/10.7554/eLife.41801.003

The following source data and figure supplement are available for figure 1:

**Source data 1.** Source data on the quantification of T antigen levels shown in *Figure 1E* and *Figure 1I*, the number of macrophages in the germband shown in *Figure 1K–M*, and the number on the yolk shown in *Figure 1—figure supplement 1N–O,Q* and on the vnc (*Figure 1—figure supplement 1P*).
DOI: https://doi.org/10.7554/eLife.41801.005

**Figure supplement 1.** Lectin screen reveals enriched staining for PNA and UEA-1 on macrophages.
DOI: https://doi.org/10.7554/eLife.41801.004

initial step of GalNAc addition to serines and threonines are numerous in *Drosophila*, with several already known to display conserved substrate specificity *in vitro* with their vertebrate orthologs (*Müller et al., 2005*; *Schwientek et al., 2002*; *Ten Hagen et al., 2003*). The *Drosophila* GalNAc-Ts affect extracellular matrix (ECM) secretion, gut acidification and the formation of the respiratory system (*Tian and Ten Hagen, 2006*; *Tran et al., 2012*; *Zhang et al., 2010*). In flies the main enzyme adding Gal to form T antigen is C1GalTA (*Müller et al., 2005*) whose absence causes defects in ventral nerve cord (vnc) condensation during Stage 17, hematopoetic stem cell maintenance, and neuromuscular junction formation (*Fuwa et al., 2015*; *Itoh et al., 2016*; *Lin et al., 2008*; *Yoshida et al., 2008*). While orthologous to the vertebrate Core1 synthases, the *Drosophila* C1GALTs differ in not requiring a specific chaperone (*Müller et al., 2005*). Most interestingly, T antigen is found on embryonic macrophages (*Yoshida et al., 2008*), a cell type which can penetrate into tissues in a manner akin to metastatic cancer (*Ratheesh et al., 2018*; *Siekhaus et al., 2010*). Macrophage invasion of the germband (*Figure 1B*, arrow in *Figure 1C*) occurs between the closely apposed ectoderm and mesoderm (*Ratheesh et al., 2018*; *Siekhaus et al., 2010*) from late Stage 11 through Stage 12. This invasion occurs as part of the dispersal of macrophages throughout the embryo (*Figure 1C*) along other routes that are mostly noninvasive, such as along the inner ventral nerve cord (vnc) (arrowhead in *Figure 1C*) (*Campos-Ortega and Hartenstein, 1997*; *Evans et al., 2010*). Given these potentially related but previously unconsolidated observations, we sought to determine the relationship between the appearance of T antigen and macrophage invasion and to use the genetic power of *Drosophila* to find new pathways by which this glycophenotype is regulated.

## Results

### T antigen is enriched and required in invading macrophages in *Drosophila* embryos

To identify glycan structures present on fly embryonic macrophages during invasion we performed a screen examining FITC-labelled lectins (see Materials and methods for abbreviations). Only two

lectins had higher staining on macrophages than on surrounding tissues (labeled enriched): PNA, which primarily binds to the core1 T O-glycan, and UEA-I, which can recognize Fucα1-2Galβ1-4GlcNAc(*Molin et al., 1986*; *Natchiar et al., 2007*) (*Figure 1D*, *Figure 1—figure supplement 1A–B*). Both glycans are associated with the invasive migration of mammalian cancer cells (*Agrawal et al., 2017*; *Hung et al., 2014*). SBA, WGA, GS-II, GS-I, ConA, MPA and BPA bound at similar or lower levels on *Drosophila* macrophages compared to flanking tissues (*Figure 1D*, *Figure 1—figure supplement 1C–I*). We saw no staining with the sialic acid-recognizing lectin LPA, and none with DBA and HPA, that both recognize α-GalNAc (*Piller et al., 1990*) (*Figure 1D*, *Figure 1—figure supplement 1J–L*). Thus PNA and UEA-I display enriched macrophage binding during their embryonic invasive migration.

To confirm T antigen as the source of the upregulated PNA signal in embryonic macrophages during invasion and to characterize its temporal and spatial enrichment, we used a monoclonal antibody (mAb 3C9) to the T O-glycan structure (*Steentoft et al., 2011*). Through Stage 10, macrophages displayed very little T antigen staining, similar to other tissues (*Figure 1E,F*). However, at late Stage 11 (*Figure 1—figure supplement 1A*) and early Stage 12, when macrophages start to invade the extended germband, T antigen staining began to be enriched on macrophages moving towards and into the germband (*Figure 1E–H*). Our results are consistent with findings showing T antigen expression in a macrophage-like pattern in late Stage 12 embryos, and on a subset of macrophages at Stage 16 (*Yoshida et al., 2008*). We knocked down the core1 synthase C1GalTA required for the final step of T antigen synthesis (*Figure 1A*) (*Lin et al., 2008*; *Müller et al., 2005*) using RNAi expression only in macrophages and observed strongly reduced staining (*Figure 1I*, *Figure 1—figure supplement 1M*). We conclude that the antibody staining is the result of T antigen produced by macrophages themselves.

To determine if these T O-glycans on macrophages are important for facilitating their germband invasion, we knocked down C1GalTA in macrophages with the RNAi line utilized above as well as one other and used the P element excision allele *C1GalTA[2.1]* which removes conserved sequence motifs required for activity (*Lin et al., 2008*). We visualized macrophages through specific expression of fluorescent markers and observed a 25 and a 33% decrease in their number in the germband for the RNAis (*Figure 1J,K*), and a 44% decrease in the C1GalTA[2.1] mutant (*Figure 1L*). When we counted the number of macrophages sitting on the yolk next to the germband in the strongest RNAi we observed an increase (*Figure 1—figure supplement 1N*) that we also observed in the C1GalT mutant (*Figure 1—figure supplement 1O*). The sum of the macrophages in the yolk and germband is the same in the control, RNAi knockdown (control 136.5 ± 6.4, RNAi 142.3 ± 6.6, p=0.7) and mutant (control 138.5 ± 4.9, mutant, 142.3 ± 7.4, p=0.87) arguing that macrophages in which C1GalTA levels are reduced cannot enter the germband but are retained on the yolk. We observed no effect on the migration of macrophages on the vnc, a route that does not require tissue invasion (*Figure 1—figure supplement 1P*) (*Campos-Ortega and Hartenstein, 1997*; *Evans et al., 2010*). 18% of T antigen in the embryo has been found to be further modified, predominantly by glucuronic acid (GlcA) (*Aoki et al., 2008*). Of the three GlcA transferases found in *Drosophila* only GlcAT-P is robustly capable of adding GlcA onto the T O-glycan structure in cells (*Breloy et al., 2016*; *Itoh et al., 2018*; *Kim et al., 2003*). To examine if the specific defect in germband invasion that we observed by blocking the formation of T antigen is due to the need for a further elaboration by GlcA, we utilized a lethal MI{MIC} transposon insertion mutant in the *GlcAT-P* gene. We observed no change in the numbers of macrophages within the germband in the *GlcAT-P[MI05251]* mutant (*Figure 1M*) and a 20% increase in the number of macrophages on the yolk (*Figure 1—figure supplement 1Q*). Therefore, our results strongly suggest that the T antigen we observe being upregulated in macrophages as they move towards and into the germband is itself needed for efficient tissue invasion.

## An atypical MFS member acts in macrophages to increase T antigen levels

We sought to determine which proteins could temporally regulate the increase in the appearance of T O-glycans in invading macrophages. We first considered proteins required for synthesizing the core1 structure, namely the T synthase, C1GalTA, and the UDP-Gal sugar transporter, Ugalt (*Aumiller and Jarvis, 2002*) (*Figure 1A*). However, q-PCR analysis of FACS sorted macrophages from Stage 9–10, Stage 12, and Stage 13–17 show that though both are enriched in macrophages,

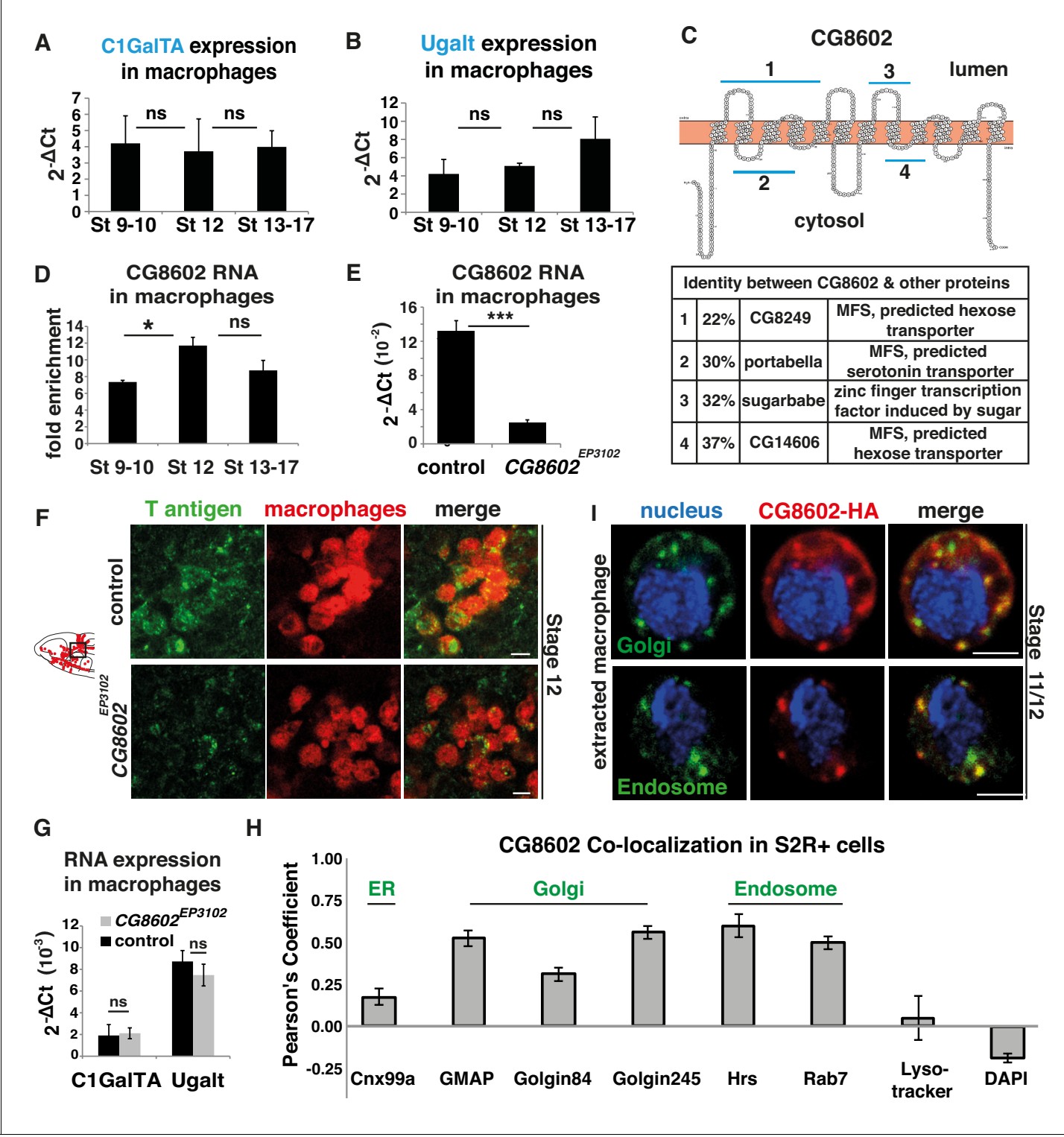

**Figure 2.** An atypical MFS family member, CG8602, located in the Golgi and endosomes, is required for T antigen enrichment on invading macrophages. (A,B) qPCR quantification ($2^{-\Delta Ct}$) of RNA levels in *mCherry+* macrophages FACS sorted from *srpHemo-3xmCherry* wild type embryos reveals no significant change in the expression of (A) the C1GalTA galactose transferase or (B) the Ugalt Gal transporter during Stage 9–17 (n = 7 biological replicates, three independent FACS sorts). (C) Schematic made with Protter (*Omasits et al., 2014*) showing the predicted 12 transmembrane domains of CG8602. Blue lines indicate regions displaying higher than 20% identity to the correspondingly numbered *Drosophila* protein indicated below, along with the homologous protein's predicted or determined function. (D) Quantification by qPCR of CG8602 RNA levels in FACS sorted *mCherry+* macrophages compared to other *mCherry-* cells obtained from *srpHemo-3xmCherry* wild type embryos at Stage 9–10, Stage 12 and Stage

*Figure 2 continued on next page*

*Figure 2 continued*

13–17. CG8602 macrophage expression peaks at Stage 12, during macrophage germband entry (n = 3–7 biological replicates, four independent FACS sorts, p=0.036). (E) qPCR quantification in FACS sorted *srpHemo-3xmCherry* labeled macrophages from control and *CG8602*<sup>EP3102</sup> mutant Stage 12 embryos shows an extremely strong decrease in CG8602 RNA expression in the P element insertion mutant used in this study (n = 7 biological replicates, three independent FACS sorts, p=0.0024). (F) Confocal images of Stage 12 control and *CG8602*<sup>EP3102</sup> mutant embryos with macrophages (red) visualized by *srpHemo-mCherry* expression and T antigen by antibody staining (green). Schematic at left depicts macrophages (red) entering the germband. Black box indicates the region next to the germband imaged at right. We observe decreased T antigen staining on macrophages in the *CG8602*<sup>EP3102</sup> mutant compared to the control. (G) qPCR quantification ($2^{-\Delta Ct}$) of C1GalTA and Ugalt RNA levels in FACS sorted macrophages from Stage 12 embryos from control and *CG8602*<sup>EP3102</sup> mutant embryos shows no significant change in expression of the Gal transferase, or the Gal and GalNAc transporter in the mutant compared to the control (n = 7 biological replicates, three independent FACS sorts). (H) Quantitation using Fiji of the colocalization of transfected *MT-CG8602::FLAG::HA* in fixed S2R+ cells with markers for the ER (Cnx99a), the Golgi (Golgin 84, Golgin 245, and GMAP), the early endosome (Hrs), the late endosome (Rab7), and live S2R+ cells transfected with *srp-CG8602::3xmCherry* with dyes that mark the lysosome (Lysotracker) and the nucleus (DAPI). Representative images are shown in ***Figure 2—figure supplement 1 B-J***. n = 24, 23, 23, 17, 6, 22, 6 and 13 cells analyzed per respective marker. (I) Macrophages near the germband extracted from *srpHemo >CG8602* HA Stage 11/12 embryos show partial colocalization of the HA antibody labeling CG8602 (red) and a Golgin 84 or Hrs antibody marking the Golgi or endosome respectively (green). Nucleus is stained by DAPI (blue). For all qPCR experiments values are normalized to expression of a housekeeping gene RpL32. Scale bars are 5 µm in **F**, 3 µm in **I**. Significance was assessed by Kruskal-Wallis test in **A, B**, One way Anova in **D** and Student's t-test in **E, G**. ns = p > 0.05, *p<0.05, ***p<0.001. See also ***Figure 2—figure supplement 1***.

DOI: https://doi.org/10.7554/eLife.41801.006

The following source data and figure supplement are available for figure 2:

**Source data 1.** Source data on the quantification of C1GalTA, Ugalt, and CG8602 expression in macrophages by qPCR (shown in ***Figure 2A–B,D–E,G***) and the Pearson's coefficient for CG8602 colocalization with different markers (***Figure 2H***).

DOI: https://doi.org/10.7554/eLife.41801.008

**Figure supplement 1.** CG8602 expression and localization.

DOI: https://doi.org/10.7554/eLife.41801.007

neither is transcriptionally upregulated before or during Stage 12 (***Figure 2A,B***). We therefore examined the Bloomington *Drosophila* Genome Project (BDGP) in situ database looking for predicted sugar binding proteins expressed in macrophages with similar timing to the observed T antigen increase (***Tomancak et al., 2007***; ***Tomancak et al., 2002***). We identified CG8602, a predicted member of the Major Facilitator Superfamily (MFS), a protein group defined by shared structural features, whose members are known to transport a diverse set of molecules across membranes (***Yan, 2015***). CG8602 contains regions of homology to known sugar responsive proteins and predicted sugar or neurotransmitter transporters (***Figure 2C***) and in a phylogenetic analysis is on a branch neighboring the SLC29 group shown to be involved in nucleoside transport (***Baldwin et al., 2004***; ***Perland et al., 2017***). BDGP in situ hybridizations (***Tomancak et al., 2007***; ***Tomancak et al., 2002***) (http://insitu. fruitfly.org/cgi-bin/ex/report.pl?ftype=10&ftext=FBgn0035763) indicate that CG8602 RNA is maternally deposited, with expression throughout the embryo through Stage 4 after which its levels decrease, with weak ubiquitous expression continuing through Stage 9–10. This is followed by strong enrichment in macrophages from Stage 11–12, with apparently equivalent levels of expression in macrophages entering the germband as in those migrating along other routes such as the ventral nerve cord. We confirmed this by q-PCR analysis of FACS sorted macrophages, which detected seven-fold higher levels of CG8602 RNA in macrophages than in the rest of the embryo by Stage 9–10 and 12-fold by Stage 12 (***Figure 2D***). These data show that RNA expression of CG8602, an MFS protein with homology to sugar transporters, increases in macrophages preceding and during the period of invasion.

To determine if CG8602 could affect T antigen levels, we examined a viable P-element insertion mutant in the 5'UTR, *CG8602*<sup>EP3102</sup> (***Figure 2—figure supplement 1A***). This insertion displays strongly reduced CG8602 expression in FACS-sorted macrophages to 15% of wild type levels, as assessed by q-PCR (***Figure 2E***). We also created an excision allele, Δ33, removing the 5'UTR flanking the P-element, the start methionine, and 914 bp of the ORF (***Figure 2—figure supplement 1A***). This is a lethal allele, and the line carrying it over a balancer is very weak; exceedingly few embryos are laid and the embryos homozygous for the mutation do not develop past Stage 12. Therefore, we did not continue experiments with this allele, and instead utilized the insertion mutant. This *CG8602*<sup>EP3102</sup> P-element mutant displays decreased T antigen staining on macrophages moving toward and entering the germband (***Figure 2F***) in Stage 11 through late Stage 12. q-PCR analysis on

FACS sorted macrophages show that the reduction in T antigen levels in the mutant is not caused by changes in the RNA levels of the T synthase C1GalTA or the Ugalt Gal and GalNAc transporter (*Aumiller and Jarvis, 2002*; *Segawa et al., 2002*) (*Figure 2G*). These results argue that CG8602 is required for enriched T antigen levels on macrophages.

To assess if CG8602 could directly regulate T antigen addition, we examined if it is found in the Golgi where O-glycosylation is initiated. We first utilized the macrophage-like S2R+ cell line, transfecting a FLAG::HA or 3xmCherry labeled form of CG8602 under the control of *srpHemo* or the copper inducible MT promoter. We detected significant colocalization with markers for the cis-Golgi marker GMAP, the Trans Golgi Network marker Golgin 245 and the endosome markers Rab7, Rab11 and Hrs (*Riedel et al., 2016*) (*Figure 2H*, *Figure 2—figure supplement 1C–G*). We detected no colocalization with markers for the nucleus, ER, peroxisomes, mitochondria or lysosomes (*Figure 2H*, *Figure 2—figure supplement 1B,H–J*). We confirmed the presence of CG8602 in the Golgi and endosomes in macrophages from late Stage 11 embryos through colocalization with Golgin 84 and Hrs, using cells extracted from positions in the head adjacent to the germband (*Figure 2I*). We conclude that the T antigen enrichment on macrophages migrating towards and into the germband requires a previously uncharacterized atypical MFS with homology to sugar binding proteins that is localized predominantly to the Golgi and endosomes.

## The MFS, Minerva, is required in macrophages for dissemination and germband invasion

We examined if CG8602 affects macrophage invasive migration. The $CG8602^{EP3102}$ mutant displayed a 35% reduction in macrophages within the germband at early Stage 12 compared to the control (*Figure 3A–B,D*, *Figure 3—figure supplement 1A*). The same decrease is observed when the mutant is placed over the deficiency Df(3L)BSC117 that removes the gene entirely (*Figure 3D*), arguing that $CG8602^{EP3102}$ is a genetic null for macrophage germband invasion. The P element transposon insertion itself causes the migration defect because its precise excision restored the number of macrophages in the germband to wild type levels (*Figure 3D*). Expression of the CG8602 gene in macrophages can rescue the $CG8602^{EP3102}$ P element mutant (*Figure 3C–D*, *Figure 3—figure supplement 1A*), and RNAi knockdown of CG8602 in macrophages can recapitulate the mutant phenotype (*Figure 3E*, *Figure 3—figure supplement 1B*). Our data thus argue that CG8602 is required in macrophages themselves for germband invasion.

Decreased numbers of macrophages in the extended germband could be caused by specific problems entering this region, or by general migratory defects or a decreased total number of macrophages. To examine the migratory step that precedes germband entry, we counted the number of macrophages sitting on the yolk next to the germband in fixed embryos in the $CG8602^{EP3102}$ mutant. We observed a 30% decrease compared to the control (*Figure 3F*), suggesting a defect in early dissemination. Entry into the germband by macrophages occurs between the closely apposed DE-Cadherin expressing ectoderm and the mesoderm and is accompanied by deformation of the ectodermal cells (*Ratheesh et al., 2018*). We tested if reductions in DE-Cadherin could ameliorate the germband phenotype. Indeed, combining the $CG8602^{EP3102}$ mutation with $shg^{P34}$ which reduces DE-Cadherin expression (*Pacquelet and Rørth, 2005*; *Tepass et al., 1996*) produced a partial rescue (*Figure 3G*), consistent with CG8602 playing a role in germband entry as well as in an earlier migratory step. There was no significant difference in the number of macrophages migrating along the vnc in late Stage 12 compared to the control in fixed embryos (*Figure 3H*) from the $CG8602^{EP3102}$ mutant or from a knockdown in macrophages of $CG8602$ by RNAi (*Figure 3—figure supplement 1C*), arguing against a general migratory defect. There was also no significant difference in the total number of macrophages in either case (*Figure 3—figure supplement 1D–E*). From analyzing the *CG8602* mutant phenotype in fixed embryos we conclude that CG8602 does not affect later vnc migration but is important for the early steps of dissemination and germband invasion.

To examine the effect of CG8602 on macrophage speed and dynamics, we performed live imaging of macrophages labeled with the nuclear marker *srpHemo-H2A::3xmCherry* in control and $CG8602^{EP3102}$ mutant embryos (*Figure 3—video 1* and *2*). We first imaged macrophages migrating from their initial position in the delaminated mesoderm up to the germband and detected a 33% decrease in speed (2.46 ± 0.07 µm/min in the control, 1.66 ± 0.08 µm/min in the $CG8602^{EP3102}$ mutant, p=0.002) (*Figure 3I,J*) and no significant decrease in persistence (0.43 ± 0.02 in the control, 0.40 ± 0.01 in the mutant, p=0.22) (*Figure 3—figure supplement 1F*). We then examined the initial

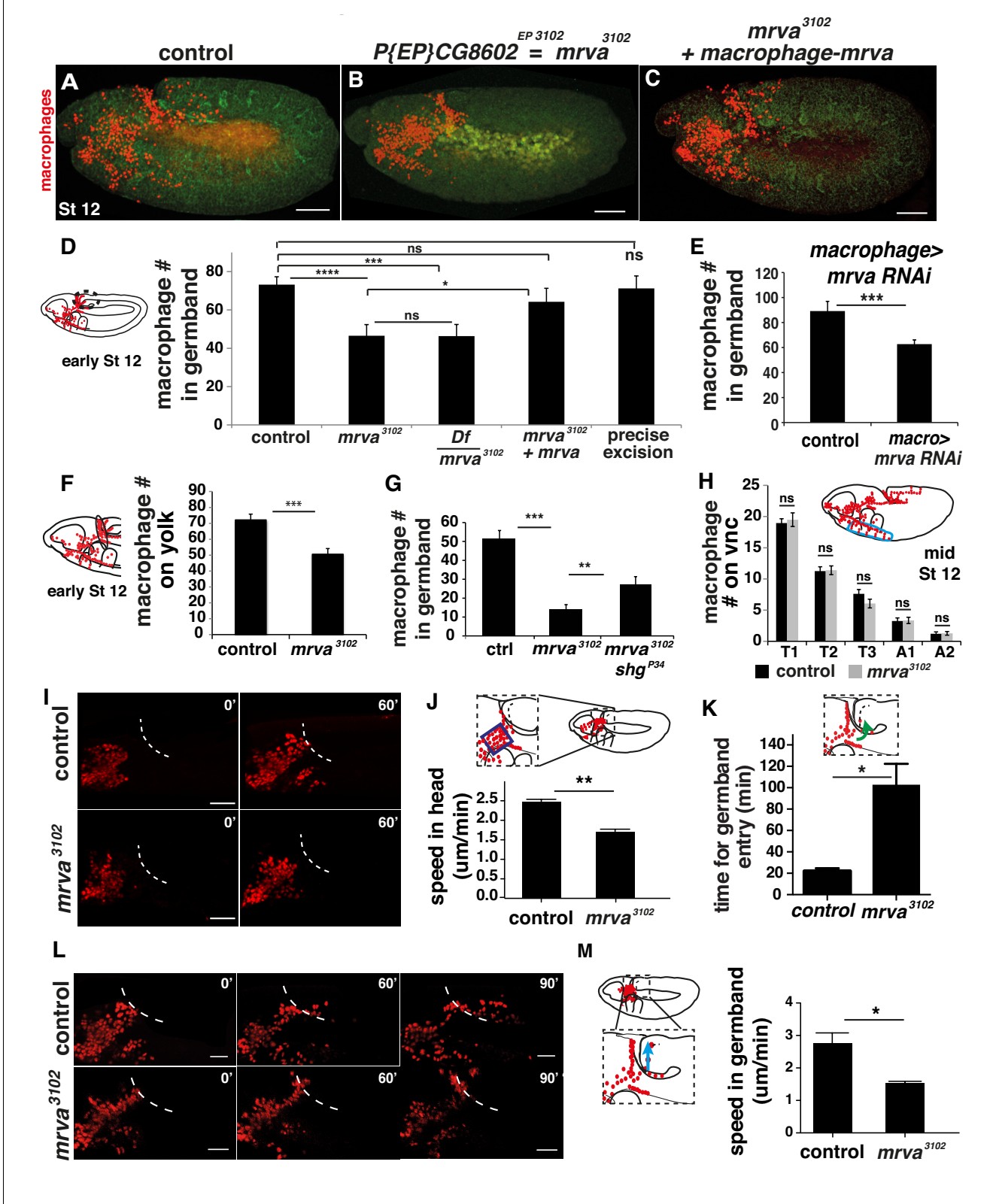

**Figure 3.** CG8602, which we name Minerva, is required in macrophages for their efficient invasion of the germband. (A–C) Representative confocal images of early Stage 12 embryos from (A) control, (B) *P{EP}CG8602^EP3102 = minerva (mrva)^3102* mutant, and (C) *mrva^3102* mutants with macrophage expression of the gene rescued by *srpHemo(macro)-mrva*. Macrophages express *srpHemo-3XmCherry* (red) and the embryo autofluoresces (green). In the mutant, macrophages remain in the head and fail to enter the germband, hence we name the gene *minerva*. (D) Dashed ellipse in schematic at left

*Figure 3 continued on next page*

*Figure 3 continued*

represents the germband region in which macrophages (red) were counted throughout the study. Comparison of the control (n= 38), *mrva^3102* mutants (n = 37) and *mrva^3102* mutant/Df(3L)BSC117 that removes the gene (n = 23) shows that the mutant significantly decreases migration into the extended germband (p<0.0001 for control vs mutant, p=-0002 for control vs Df cross). This defect can be partially rescued by expression in macrophages of *srpHemo >mrva::FLAG::HA* (n = 18, p=0.222 for control vs rescue, p=0.036 for mutant vs rescue) and completely rescued by precise excision (*mrva^Δ32*) of the P element (n = 16, p=0.826). *srpHemo >mCherry* nls labeled the macrophages. (**E–G**) Macrophage quantification in early Stage 12 embryos. (**E**) Fewer germband macrophages upon expression of *mrva* RNAi *v101575* only in macrophages under the control of *srpHemo* (n = 28–35 embryos, p<0.0001). (**F**) Fewer macrophages found on the yolk neighboring the germband (oval in schematic) in the *mrva^3102* mutant compared to control embryos (n = 14–16 embryos, p=0.0003). (**G**) Increased germband macrophage numbers in *shg^P34*; *mrva^3102* compared to the *mrva^3102* mutant indicates a partial rescue from reducing DE-Cadherin which is expressed in the germband ectoderm (n = 19–29, p<0.0001, p=0.005). (**H**) No significant difference in number of macrophages labeled with *srpHemo-3xmCherry* in vnc segments (area in blue oval in schematic) between control and *mrva^3102* mutant embryos in fixed mid Stage 12 embryos (n = 23–25, p=0.55). Images from two-photon movies of (**I**) Stage 10 and (**L**) late Stage 11-early Stage 12 embryos in which macrophage nuclei (red) are labeled with *srpHemo-H2A::3xmCherry*. (**I**) Stills at 0 and 60 min and (**J**) quantification of macrophage speed reveal 33% slower macrophage migration in the head towards the yolk neighboring the germband in the *mrva^3102* mutant compared to the control, n = 3 movies for each, #tracks: control = 329, mutant = 340, p=0.002. Blue box in magnification in schematic indicates region analysed in J. (**K**) The first macrophage in *mrva^3102* mutants is much slower to enter the germband after macrophages reach the germband edge (control = 22.00 ± 1.53 min, n = 3, *mrva^3102* mutant = 102.0 ± 20.35 min, n = 4. p-value=0.021). (**L**) The time when macrophages reached the germband in each genotype was defined as 0'. Stills at 60 and 90 min and (**M**) quantification of macrophage speed reveal 43% slower macrophage migration in the germband in the *mrva^3102* mutant compared to the control. Blue arrow in schematic indicates route analyzed. n = 3 movies for each, #tracks: control = 21, mutant = 14, p=0.022. Significance was assessed by Kruskal-Wallis test with Conover post test comparison in D, G, Student's t-test in E, F, H, J-K, M. ns = p > 0.05, *p<0.05, **p<0.01, ***p<0.001, ***p<0.0001. Scale bars are 50 μm in A-C, 40 μm in I, 30 μm in L. See also *Figure 3—figure supplement 1* and *Figure 3—video 1–3*.

DOI: https://doi.org/10.7554/eLife.41801.009

The following video, source data, and figure supplement are available for figure 3:

**Source data 1.** Source data on the quantification of macrophages in the germband shown in *Figure 3D–E,G* and *Figure 3—figure supplement 1A*, on the yolk (*Figure 3F*) on the vnc (*Figure 3H*, *Figure 3—figure supplement 1C*) and in the whole embryo (*Figure 3—figure supplement 1D–E*).

DOI: https://doi.org/10.7554/eLife.41801.011

**Figure supplement 1.** CG8602 (Minerva) affects macrophage migration into the germband but not along the vnc and does not alter border cell or germ cell migration.

DOI: https://doi.org/10.7554/eLife.41801.010

**Figure 3—video 1.** Representative movie of macrophage migration into the germband in the control.

DOI: https://doi.org/10.7554/eLife.41801.012

**Figure 3—video 2.** Representative movie of macrophage migration into the germband in the *mrva^3102* mutant.

DOI: https://doi.org/10.7554/eLife.41801.013

**Figure 3—video 3.** Representative movies of macrophage migration on the vnc in the control and *mrva^3102* mutant.

DOI: https://doi.org/10.7554/eLife.41801.014

migration of macrophages into the germband at late Stage 11. We observed a range of phenotypes in the six movies we made of the mutant, with macrophages pausing at the germband edge from twice to six times as long as in the control before invading into the tissue (*Figure 3K* shows average time for entry, control = 22.00 ± 1.53 min, *CG8602^EP3102* mutant = 102.0 ± 20.35 min). As we observed no change in the timing of the initiation of germband retraction (269.6 ± 9 min in control and 267.1 ± 3 min in mutant, p=0.75) but did observe a decreased speed of its completion in the mutant (107 ± 12 min from start to end of retraction in control and 133 ± 6 min for mutant p=0.05), we only analyzed macrophages within the germband before its retraction begins. We observed a 43% reduction in macrophage speed within the germband (2.72 ± 0.32 μm/min in the control and 1.55 ± 0.04 μm/min in the mutant, p=0.02) (*Figure 3L,M*). To assess this phenotype's specificity for invasion, we used live imaging of macrophage migration along the inner vnc that occurs during the same time period as germband entry; we observed no significant change in speed (2.41 ± 0.06 μm/ min in the control and 2.23 ± 0.01 μm/min in the mutant, p=0.11) or directionality (0.43 ± 0.03 in the control and 0.43 ± 0.02 in the mutant, p=0.9742) (*Figure 3—figure supplement 1G*, *Figure 3— video 3*). We conclude from the sum of our experiments in fixed and live embryos that CG8602 is important for the initial disseminatory migration out of the head and for invasive migration into and within the germband, but does not alter general migration. We name the gene *minerva (mrva)*, for the Roman goddess who was initially trapped in the head of her father, Jupiter, after he swallowed her pregnant mother who had turned herself into a fly.

## Minerva is not required for border cell invasion or germ cell migration

To assess if Minerva only affects macrophage invasion or also other types of tissue penetration in *Drosophila*, we examined the migration of germ cells and border cells. Germ cells move in an Integrin-independent fashion through gaps in the midgut created by ingressing formerly epithelial cells (*Devenport and Brown, 2004*; *Seifert and Lehmann, 2012*). We found no defect in germ cell migration when examining control and *mrva³¹⁰²* mutant embryos stained with the Vasa Ab (*Figure 3—figure supplement 1H–I*). Border cells are born in the epithelia surrounding the ovary and then delaminate to move invasively between the nurse cells towards the oocyte (*Montell, 2003*), guided by the same receptor that macrophages use during their embryonic dispersal, PVR (*Duchek et al., 2001*). They migrate as a tumbling collective, using invadopodia and Cadherin-based adhesion to progress (*Cai et al., 2014*; *Niewiadomska et al., 1999*). *mrva* is expressed in dissected control ovaries and the *mrva³¹⁰²* mutant reduces the levels of *mrva* RNA in the ovary by 70%, similar to the reduction observed in macrophages (*Figure 3—figure supplement 1J*). We identified border cells by staining with DAPI to detect their clustered nuclei. We observed no significant change in border cell migration towards the oocyte in the *mrva³¹⁰²* mutant compared to the control (*Figure 3—figure supplement 1K–L*). These results support the conclusion that Mrva is not generally required for all migratory cells that move confined through tissues during development, but specifically for the invasion of macrophages, which is an Integrin-dependent process (*Siekhaus et al., 2010*).

## Minerva affects a small fraction of the *Drosophila* embryonic O-glycoproteome

We set out to determine if Minerva induces T glycoforms on particular proteins. We first conducted a Western Blot with a mAb to T antigen on whole embryo extracts. We used the whole embryo because we were unable to obtain enough protein from FACS sorted macrophages or to isolate CRISPR-induced full knockouts of *minerva* in the S2R+ macrophage like cell line. We observed that several bands detected with the anti-T mAb were absent or reduced in the *minerva* mutant (*Figure 4A*), indicating an effect on the T antigen modification of a subset of proteins.

We wished to obtain a more comprehensive view of the proteins affected by Minerva. Since there is little information about *Drosophila* O-glycoproteins and O-glycosites (*Schwientek et al., 2007*; *Aoki and Tiemeyer, 2010*), we used lectin-enriched O-glycoproteomics to identify proteins displaying T and Tn glycoforms in Stage 11/12 embryos from wild type and *mrva³¹⁰²* mutants (*Figure 4—figure supplement 1A*). We labeled tryptic digests of embryonic protein extracts from control or mutant embryos with stable dimethyl groups carrying medium ($C_2H_2D_4$) or light ($C_2H_6$) isotopes respectively to allow each genotype to be identified in mixed samples (*Boersema et al., 2009*; *Schjoldager et al., 2012*; *Schjoldager et al., 2015*). The pooled extracts were passed over a Jacalin column to enrich for T and Tn O-glycopeptides; the eluate was analyzed by mass spectrometry to identify and quantify T and Tn modified glycopeptides in the wild type and the mutant sample through a comparison of the ratio of the light and medium isotope labeling channels for each glycopeptide (see *Figure 4—figure supplement 1B–C* for example spectra).

In the wild type we identified T and Tn glycopeptides at 936 glycosites derived from 270 proteins (*Supplementary file 1* and *Figure 4B*). 62% of the identified O-glycoproteins and 77% of identified glycosites contained only Tn O-glycans. 33% of the identified O-glycoproteins and 23% of glycosites displayed a mixture of T or Tn O-glycans, and 5% of identified O-glycoproteins and 4% of glycosites had solely T O-glycans (*Figure 4C*). In agreement with previous studies (*Steentoft et al., 2013*), only one glycosite was found in most of the identified O-glycoproteins (44%) (*Figure 4D*). In 20% we found two sites, and some glycoproteins had up to 27 glycosites. The identified O-glycosites were mainly on threonine residues, (78.5%) with some on serines (21.2%) and very few on tyrosines (0.3%) (*Figure 4—figure supplement 1D*). Metabolism, cuticle development, and receptors were the most common functional assignments for the glycoproteins (*Figure 4—figure supplement 1E*).

We sought to assess the changes in glycosylation in the *mrva* mutant. A majority of the quantifiable Tn and T O-glycoproteome was unaltered between the wild type and the *mrva³¹⁰²* mutant, with only 63 proteins (23%) showing more than a three-fold change and 18 (6%) a ten-fold shift (*Figure 4F*). We observed both increases and decreases in the levels of T and Tn modification on proteins in the mutant (*Figure 4F–G*, *Supplementary file 1* and *2*), but a greater number of proteins showed decreased rather than increased T antigen levels. 67% of the vertebrate orthologs of

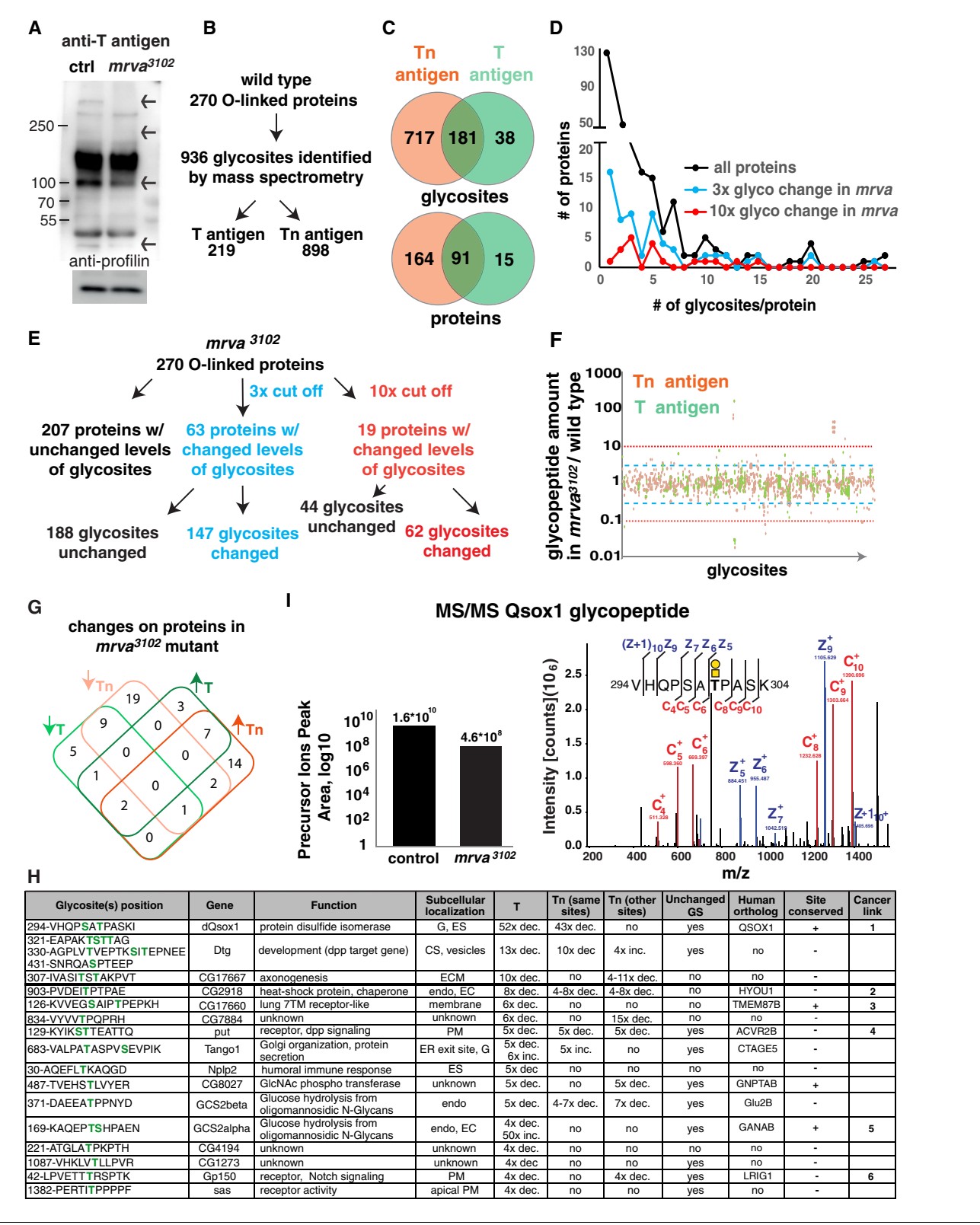

**Figure 4.** Glycoproteomic analysis reveals Minerva is required for higher levels of T-antigen on a subset of proteins. (A) Representative Western blot of protein extracts from Stage 11/12 control and *mrva³¹⁰²* mutant embryos probed with T antigen antibody. Arrows indicate decreased/missing bands in the mutant compared to the control. Profilin serves as a loading control (n = 10 biological replicates). (B) Summary of glycomics results on wild type embryos. (C) Venn diagram indicating number of glycosites or proteins found with T, Tn or T and Tn antigen modifications in the wild type. (D) Plot

Figure 4 continued

showing the number of T and Tn antigen glycosites per protein in the total glycoproteome and on proteins that show three (blue) and ten-fold (red) altered glycopeptides in the $mrva^{3102}$ mutant. Proteins strongly affected by Minerva have a higher number of glycosites (p=0.005). (E) Summary of glycomics on $mrva^{3102}$ embryos showing the numbers of proteins and glycosites exhibiting three (blue) or ten (red) fold changes in T and Tn antigen levels. (F) T antigen (in green) and Tn antigen (orange) occupied glycosites plotted against the ratio of the levels of glycopeptides found for each glycosite in the $mrva^{3102}$ mutant/control. Higher positions on the plot indicate a lower level of glycosylation in the mutant. Blue dashed line represents the cut off for 3x changes in glycosylation, and the red dotted line the 10x one. (G) Venn diagram of the number of proteins with at least three fold change in the T antigen (T, green) or Tn antigen (Tn, orange) glycosylation in the $mrva^{3102}$ mutant. Up arrows denote increase, down arrows indicate decrease in levels. (H) Proteins with at least a three fold decrease in T antigen levels in the $mrva^{3102}$ mutant. Glycan modified amino acids are highlighted in bold green font. Unchanged/Higher GS column indicates if any other glycosite on the protein is unchanged or increased. Table does not show the two chitin and chorion related genes unlikely to function in macrophages. G: Golgi, ES: Extracellular space, Endo: Endosomes, ER: Endoplasmic reticulum, ECM: Extracellular Matrix, PM: Plasma Membrane, GS: Glycosite. Cancer links as follows. 1) QSOX1: Promotes cancer invasion in vitro, overexpression worse patient outcomes (*Katchman et al., 2013*; *Katchman et al., 2011*). 2) HYOU1: Overexpression associated with vascular invasion, worse patient outcomes (*Stojadinovic et al., 2007*) (*Zhou et al., 2016*). 3) TMEM87B: translocation breakpoint in cancer, (*Hu et al., 2018*). 4) ACVR2B: over expressed in renal cancer (*Senanayake et al., 2012*). 5) GANAB: inhibits cancer invasion in vitro (*Chiu et al., 2011*). 6) LRIG1: inhibits cancer invasion in vitro, and in mice (*Sheu et al., 2014*), (*Mao et al., 2018*). (I) Annotated ETD MS2 spectra of the VHQPSATPASK glycopeptide from Qsox1 with T antigen glycosylation at position T7. See schematic in which the yellow square represents GalNAc and the yellow circle Gal. Assigned fragment ions in MS2 spectra are highlighted by red for 'c' type fragments (those retaining the original N terminus) and blue for 'z' type fragments (those retaining the original C terminus). The graph at the left shows the relative quantification of the glycopeptide precursor ion's peak area in the control and $mrva^{3102}$ mutant plotted on a logarithmic scale. See also *Figure 4—figure supplement 1*, *Figure 4*-Dataset 1, *Supplementary file 1–3*.
DOI: https://doi.org/10.7554/eLife.41801.016

The following figure supplement is available for figure 4:

**Figure supplement 1.** Related to *Supplementary file 1*: Further information on the mass spectrometry results.
DOI: https://doi.org/10.7554/eLife.41801.017

*Drosophila* proteins displaying shifts in this O-glycosylation have previously been linked to cancer (*Figure 4H*, *Supplementary file 2*). These proteins were affected at specific sites, with 40% of glycosites on these proteins changed more than three fold and only 14% more than ten fold. The glycosite shifts in T antigen occurred either without significant alterations in Tn (33% of glycosites had only decreased T antigen, 17% of glycosites had only increased T antigen) or with changes in T antigen occurring in the same direction as the changes in Tn (22% of glycosites both Tn and T antigen increased, 22% of glycosites both Tn and T decreased) (*Supplementary file 2*). Only 1% of glycosites displayed decreased T antigen with a significant increase in Tn. Interestingly, a higher proportion of the glycoproteins with altered O-glycosylation in the $mrva^{3102}$ mutant had multiple glycosites than the general glycoproteome (*Figure 4D*) (p value=0.005 for ten-fold changes). We conclude that Minerva affects O-glycosylation occupancy on a small subset of O-glycoproteins, many of whose vertebrate orthologs have been linked to cancer, with both T and Tn O-glycopeptides being affected.

## Minerva raises T antigen levels on proteins required for invasion

Given that blocking Tn to T conversion through the knockdown of the C1GalTA enzyme resulted in a germband invasion defect, we examined the known functions of the 18 proteins with lower T antigen in the absence of Minerva to distinguish which processes Minerva could influence to facilitate invasion (*Figure 4H*). We excluded two proteins involved in eggshell and cuticle production. To spot proteins whose reduced T antigen-containing glycopeptides are caused directly by alterations in glycosylation rather than indirectly by decreased protein expression in the *mrva* mutant, we checked if glycosylation at other identified glycosites was unchanged or increased. We identified ten such proteins, several of which were in pathways that had been previously linked to invasion in vertebrates. Qsox1, a predicted sulfhydryl oxidase required for the secretion and thus potential folding of EGF repeats (*Tien et al., 2008*) showed the strongest alterations of any protein, with a 50-fold decrease in T antigen levels in the *mrva* mutant (*Figure 4I*). The mammalian ortholog QSOX1 has been shown to affect disulfide bond formation, is overexpressed in some cancers, promotes Matrigel invasion, and can serve as a negative prognostic indicator in human cancer patients (*Chakravarthi et al., 2007*; *Katchman et al., 2011*; *Lake and Faigel, 2014*). Dtg, with a 13-fold reduction in T antigen (*Hodar et al., 2014*), and Put with a five-fold reduction (*Letsou et al., 1995*) respond to signaling by the BMP-like ligand, Dpp. Dpp signaling directs histoblast invasion in the fly (*Ninov et al., 2010*). Gp150 shows a four fold decrease in T antigen and modulates Notch signaling (*Fetchko et al.,*

*2002*; *Li et al., 2003*). Notch and BMP promote invasion and metastasis in mice (*Bach et al., 2018*; *Garcia and Kandel, 2012*; *Owens et al., 2015*; *Pickup et al., 2015*; *Sahlgren et al., 2008*; *Sonoshita et al., 2011*). We conclude that Mrva is required to increase T O-glycans on a subset of the glycosites of selected glycoproteins involved in protein folding, glycosylation and signaling in pathways frequently linked to promoting cancer metastasis. Its strongest effect is on a predicted sulfhydryl oxidase, the *Drosophila* ortholog of the mammalian cancer protein, QSOX1.

We wished to determine how Qsox1 might affect *Drosophila* macrophage germband invasion. Embryos from the KG04615 P element insertion in the 5'UTR of the *qsox1* gene displayed 42% fewer macrophages in the germband compared to the control (*Figure 5A,B*) with an increase in macrophages remaining on the yolk (*Figure 5—figure supplement 1A*). We observed a small decrease in migration along the vnc (*Figure 5—figure supplement 1B*) and no change in total macrophage numbers in these embryos (*Figure 5—figure supplement 1C*). These migration phenotypes were also observed in embryos in which RNAi line v108288 knocked down *qsox1* only in macrophages (*Figure 5C*, *Figure 5—figure supplement 1D–E*). We then conducted live imaging (*Figure 5D*, *Figure 5—video 1*) (compare to control shown in *Figure 3—video 1*) to examine how the $qsox1^{KG04615}$ mutant affected the dynamics of macrophage migration. During the movement of macrophages labeled with the nuclear marker *srpHemo-H2A::3xmCherry* from their initial position up to the germband we detected an 18% decrease in speed (*Figure 5E*) (2.46 ± 0.07 µm/min in the control, 2.02 ± 0.03 µm/min in the $qsox1^{KG046152}$ mutant, p=0.006, n = 3) and no significant decrease in persistence (*Figure 5—figure supplement 1F*) (0.43 ± 0.02 in the control, 0.39 ± 0.01 in the mutant, p=0.13). Macrophages in the *qsox1* mutant were delayed twice as long at the germband edge before entering (*Figure 5F*) (time to entry 22.00 ± 1.53 min in the control and 49.67 ± 9.33 min in the $qsox1^{KG046152}$ mutant, n = 3). Once in, they moved within the germband with a 17% slower speed, a reduction that was not statistically significant (*Figure 5G*) (2.72 ± 0.32 µm/min in the control, 2.27 ± 0.20 µm/min in the $qsox1^{KG046152}$ mutant, p=0.30, n = 3). We conclude that Qsox1 aids the disseminatory migration of macrophages but is most strongly required for their initial invasion into the germband tissues.

We wished to examine how Qsox1 could be exerting this effect on macrophage tissue entry. Vertebrate QSOX1 has been shown to localize to the Golgi and act as a sulfhydryl oxidase, catalyzing disulfide bond formation and protein folding (*Alon et al., 2012*; *Chakravarthi et al., 2007*; *Heckler et al., 2008*; *Hoober et al., 1996*). The *Drosophila* protein has been shown to be required for the secretion of multimerized EGF domains and was hypothesized to act redundantly with ER oxidoreductin-like-1 to form disulfide bonds (*Tien et al., 2008*). We found that an HA-tagged form of Qsox1 transfected into the *Drosophila* macrophage like cell line, S2R+, colocalizes little with markers for the ER, and considerably with those for Golgi and endosomes (*Figure 5H*, *Figure 5—figure supplement 1G–I*). We also observed significant colocalization with 3xmCherry-tagged Mrva (*Figure 5H*, *Figure 5—figure supplement 1J*). Vertebrate QSOX1 can be cleaved from its transmembrane domain to allow secretion (*Rudolf et al., 2013*), and has been shown *in vitro* to be required extracellularly for the incorporation of laminin produced by fibroblasts into the extracellular matrix (ECM), thereby supporting efficient cancer cell migration (*Ilani et al., 2013*). *Drosophila* Qsox1 also has a transmembrane domain, yet we detected an HA-tagged form in the media after transfection into S2R+ cells (*Figure 5I*), indicating that it can be secreted. To examine if *Drosophila* Qsox1 might also affect Laminin, we stained $mrva^{3102}$ and $qsox1^{KG046152}$ mutant embryos with an antibody against Laminin A (LanA) (*Figure 5—figure supplement 1K*). In both mutants we observed increased amounts of LanA inside and somewhat higher levels adjacent to the macrophages, but no significant alteration at the cell edges compared to the control (*Figure 5J*, *Figure 5—figure supplement 1L–N*). We conclude that *Drosophila* Qsox1 can be secreted but is also found in the Golgi and endosomes like Mrva, and that both proteins affect LanA, a component of the ECM.

## Conservation of Minerva's function in macrophage invasion and T antigen modification by its mammalian ortholog MFSD1

To determine if our studies could ultimately be relevant for mammalian biology and therefore also cancer research, we searched for a mammalian ortholog. MFSD1 from *mus musculus* shows strong sequence similarity with Mrva, with 50% of amino acids displaying identity and 68% conservation (*Figure 6A*, *Figure 6—figure supplement 1A*). A transfected C-terminally GFP-tagged form (*Figure 6—figure supplement 1B*) showed localization to the secretory pathway, colocalizing with the

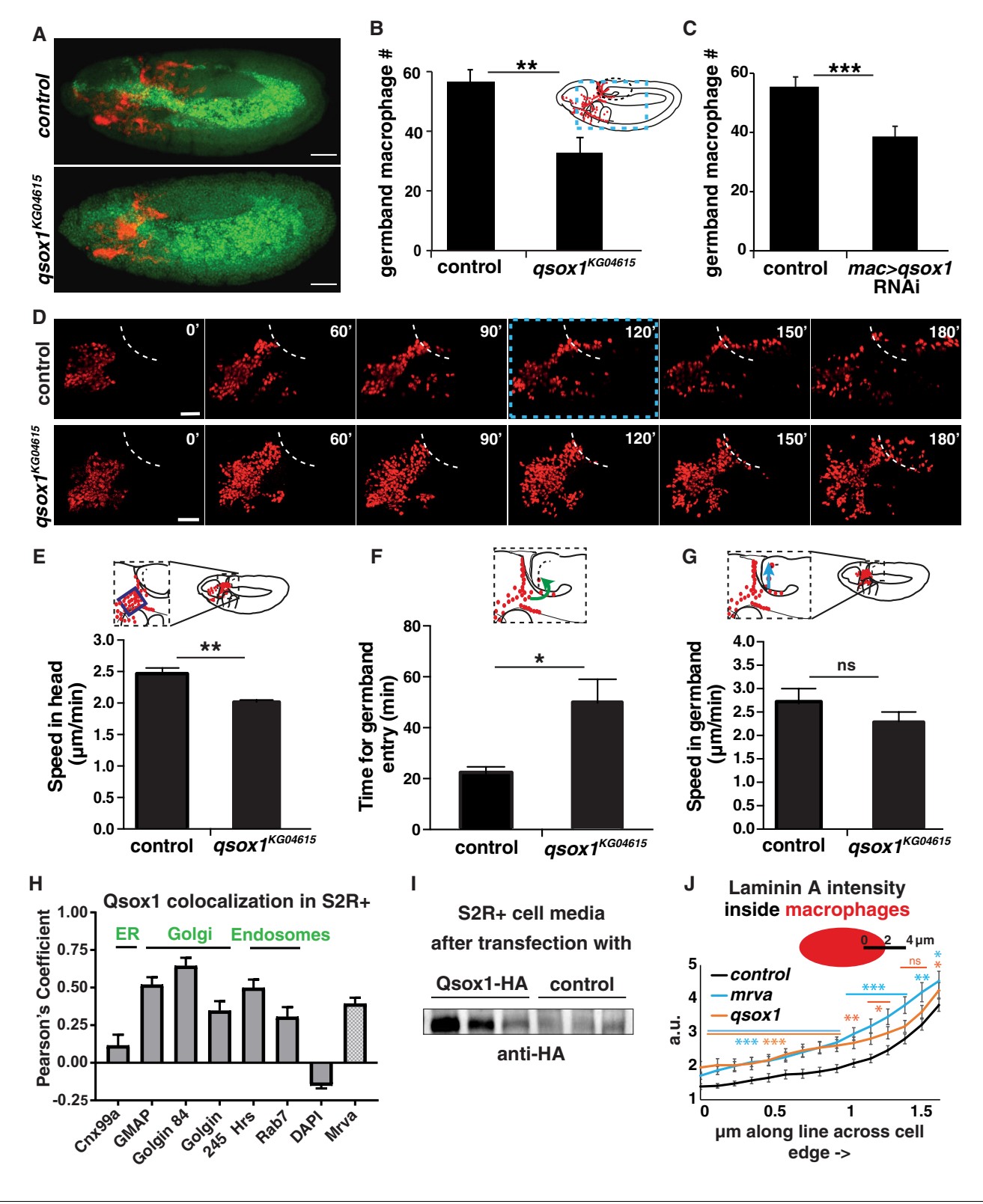

**Figure 5.** Qsox1 is required for macrophage dissemination and entry into the germband tissue. (**A**) Representative confocal images of early Stage 12 embryos from control and *P{SUPor-P}Qsox1KG04615 = qsox1^{KG04615}*. (**B–C**) Quantification in early Stage 12 embryos showing a significant reduction in germband macrophages (**B**) in the P-element mutant *qsox1^{KG04615}* located in the Qsox1 5'UTR (n = 18, p=0.0012) and (**C**) upon the expression in macrophages under *srpHemo-GAL4* control of an RNAi line (*v108288*) against Qsox1 (n = 24, 23 embryos, p=0.001). (**D**) Images from two-photon

*Figure 5 continued on next page*

*Figure 5 continued*

movies from control and *qsox1^KG04615*. Macrophage nuclei (red) are labeled with *srpHemo-H2A::3xmCherry.* Stills at 0, 60, 90, 120, 150 and 180 min. (**E**) Quantification of macrophage speed reveals 18% slower macrophage migration in the head towards the yolk neighboring the germband in the *qsox1^KG04615* mutant compared to the control (n = 3 movies for each, #tracks: control = 329, mutant = 396, p=0.0056). (**F**) Quantification of the time required for macrophage entry into the germband in *qsox1^KG04615* compared to the control. n = 3 movies for each, p=0.043. (**G**) Quantification of macrophage speed in the germband in the *qsox1^KG04615* mutant compared to the control (n = 3 movies for each, #tracks: control = 21, mutant = 19, p=0.300). (**H**) Pearson's Coefficient analysis indicating the level of colocalisation of a *MT-Qsox1::FLAG::HA* construct transfected into S2R+ cells visualized with an HA antibody and antibodies against markers for the ER (Cnx99a), the Golgi (Golgin 84, Golgin 245, and GMAP), the early endosome (Hrs), the late endosome (Rab7) and the nucleus (DAPI) (n = 11–15) as well as with a *srpHemo-mrva::3xmCherry* construct (n = 18). (**I**) Western blot of concentrated supernatant collected from S2R+ cells transfected with *srpGal4 UAS-qsox1::FLAG::HA* (first three lines) and S2R+ cells that are untransfected. (**J**) Quantification of intracellular LanA intensity along a 4 µm line in macrophages (as indicated in schematic) from the control (black), *minerva^3102* (blue) and the *qsox1^KG04615* mutants (orange) (n = 4–5 embryos, 80–100 cells, 240–300 lines). For the whole graph see *Figure 5—figure supplement 1G–J*. Scale bars 50 µm for A, 30 µm in D. B-C, E-G and J were analyzed with Student's test. ns = p > 0.05, *p<0.05, **p<0.01, ***p<0.001. See also *Figure 5—figure supplement 1*.

DOI: https://doi.org/10.7554/eLife.41801.018

The following video, source data, and figure supplement are available for figure 5:

**Source data 1.** Source data on the quantification of macrophages in the germband shown in *Figure 5B-C*, on the yolk shown in *Figure 5—figure supplement 1A,1D*, on the vnc shown in *Figure 5—figure supplement 1B,1E*, and in the whole embryo shown in *Figure 5—figure supplement 1C*.

DOI: https://doi.org/10.7554/eLife.41801.020

**Figure supplement 1.** Qsox1 affects germband entry and Laminin A.

DOI: https://doi.org/10.7554/eLife.41801.019

**Figure 5— video 1.** Representative movie of macrophage migration into the germband in the *qsox1^KG04615* mutant.

DOI: https://doi.org/10.7554/eLife.41801.021

Golgi marker GRASP65 in murine MC-38 colon carcinoma, 4T1 breast cancer cells and LLC1.1 lung cancer (*Figure 6B–C*, *Figure 6—figure supplement 1C–E*) and with Golgi and endosomal markers in B16-BL6 melanoma cells (*Figure 6C*, *Figure 6—figure supplement 1F*). mmMFSD1 expression in macrophages in *mrva^3102* mutant embryos can completely rescue the germband invasion defect (*Figure 6D–E*). This macrophage-specific expression of MFSD1 also resulted in higher levels of T antigen on macrophages when compared to those in *mrva^3102* mutants (*Figure 6F–G*). Thus MFSD1 not only displays localization in the Golgi apparatus in multiple types of mammalian cancer but can also rescue O-glycosylation and migration defects when expressed in *Drosophila*, arguing that the functions Mrva carries out to promote invasion into the germband are conserved up to mammals.

## Discussion

O-glycosylation is one of the most common posttranslational modifications, yet the intrinsic technical challenges involved in identifying O-glycosites and altered O-glycosylation on a proteome-wide level has hampered the discovery of biological functions (*Levery et al., 2015*). Here we provide two important new advances for the field. First, we identify a key regulator of this O-glycosylation, Minerva, with an unexpected role for a member of the major facilitator superfamily. Our demonstration that this conserved protein affects invasion and the appearance of the cancer-associated core1 T glycoform on a set of proteins connected to invasion provides a new perspective on T glycoform regulation and may have implications for cancer. Second, we define the GalNAc-type O-glycoproteome of *Drosophila* embryos. As O-glycosites cannot as yet be reliably predicted, our proteomic characterization in a highly genetically accessible organism will permit future studies on how glycosylation affects cell behavior; we highlight T and Tn O-glycosylated receptors in *Supplementary file 3* to further this goal.

### Modifications of the O-glycoproteome by an MFS family member

Our identification of a MFS family member as a regulator of O-glycosylation is surprising. MFS family members can serve as transporters and shuttle a wide variety of substrates (*Quistgaard et al., 2016*; *Reddy et al., 2012*). Minerva displays homology to sugar transporters and is localized to the Golgi and endosomes. Minerva could thus affect O-glycosylation in the Golgi through substrate availability. However, the lower and higher levels of glycosylation in the *mrva^3102* mutant we observe are hard to reconcile with this hypothesis. Given that the changes in T antigen on individual

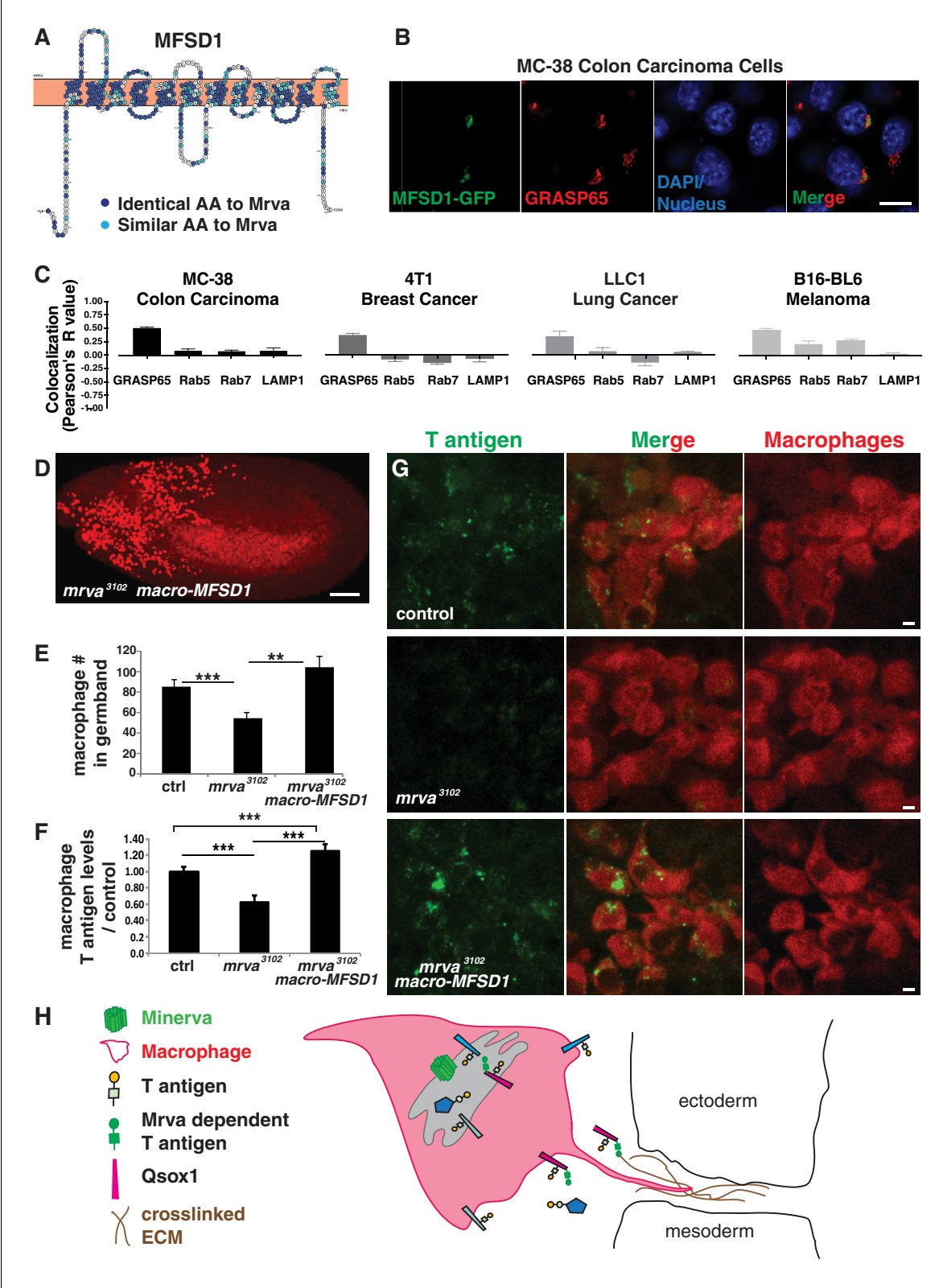

**Figure 6.** Minerva's murine ortholog, MFSD1, can substitute for Minerva's functions in migration and T-antigen glycosylation. (**A**) Topology prediction of mouse MFSD1 (NP_080089.1) using the online tools TMPred (*Hofman and Stoffel, 1993*) and Protter (*Omasits et al., 2014*). 50% of amino acids are identical between the *M. musculus* MFSD1 and *D. melanogaster* sequence of *mrva (CG8602)* (NP_648103.1) and are highlighted in dark blue, similar amino acids are in light blue. (**B**) Confocal images of MC-38 colon carcinoma cells showing colocalization of MFSD1-eGFP (green) with the Golgi marker

*Figure 6 continued on next page*

*Figure 6 continued*

GRASP65 (red). DAPI labels the nucleus (blue). (**C**) Quantitation using Fiji of the colocalization of MFSD1-eGFP with the Golgi marker (GRASP65), early endosome marker (Rab5), late endosome marker (Rab7), and lysosome marker (LAMP1) in MC-38 colon carcinoma, B16-BL6 melanoma, LLC1 Lewis lung carcinoma, and 4T1 breast carcinoma cells. Representative images are shown in *Figure 6—figure supplement 1C–F* (n = 8–15, 5–9, 4–9, 5–10 cells per condition within the respective cancer types). (**D**) Confocal image of a Stage 12 fixed embryo showing that expression of *mmMFSD1* in macrophages under the direct control of the *srpHemo(macro)* promoter in the *mrva³¹⁰²* mutant can rescue the defect in macrophage migration into the germband. Compare to *Figure 3A,B*. Macrophages visualized with *srpHemo-H2A::3xmCherry* for D-E. (**E**) Quantitation of the number of macrophages in the germband of early Stage 12 embryos from the control (n = 25), *mrva³¹⁰²* mutants (n = 29), and *mrva³¹⁰²* *srpHemo(macro)-mmMFSD1* (n = 13, p=0.0005 for mutant vs control, p<0.0001 for mutant vs rescue). (**F**) Quantification of T antigen levels on macrophages in late Stage 11 embryos from control, *mrva³¹⁰²*mutant and *mrva³¹⁰²* *srpHemo(macro)-mmMFSD1* embryos. T antigen levels normalized to those observed in the control (n = 8–9 embryos, 280, 333, and 289 cells quantified respectively, p<0.0001 for both). (**G**) Confocal images of macrophages (red) on the germband border stained with T antigen antibody (green) in the control, the *mrva³¹⁰²* mutant, and *mrva³¹⁰²* *srpHemo(macro)-mmMFSD1* shows that mmMFSD1 expression in macrophages can rescue the decrease of macrophage T antigen observed in the *mrva³¹⁰²* mutant. Macrophages visualized with *srpHemo-3xmCherry* for F-G. (**H**) Model for Minerva's function during macrophage invasion based on our findings and the literature: Minerva in the Golgi (grey) leads to increases in T antigen levels on a subset of proteins that aid invasion, including Qsox1 which regulates protein folding through disulfide bond formation and isomerization. We propose that increased T antigen on Qsox1 facilitates its sulfhydryl oxidase activity that aids the formation of a robust crosslinked ECM which macrophages utilize during tissue entry. Significance was assessed by Kruskal-Wallis test with Conover post test analysis in E,F). ***p<0.001, ****p<0.0001. Scale bars are 10 µm in B, 50 µm in D, and 3 µm in G. See also *Figure 6—figure supplement 1*.
DOI: https://doi.org/10.7554/eLife.41801.022

The following source data and figure supplement are available for figure 6:

**Source data 1.** Source data on the quantification of Pearson's coefficient for MFSD1 colocalization with different markers (*Figure 6C*), the number of macrophages in the germband (*Figure 6E*) and the level of T antigen in macrophages (*Figure 6F*).
DOI: https://doi.org/10.7554/eLife.41801.024
**Figure supplement 1.** MFSD1-eGFP localization in colon, breast, lung and skin cancer cells.
DOI: https://doi.org/10.7554/eLife.41801.023

glycosites in the *mrva* mutant are found either with no significant change in Tn or with a change in the same direction (*Supplementary file 1* and *2*), regulation appears to occur at the initial GalNAc addition on the protein subset as well as on further T antigen elaboration. 95% of the proteins with 10-fold altered glycosylation in the *mrva* mutant had multiple O-glycosylation sugar modifications compared to 56% of the general O-glycoproteome. Greatly enhanced glycosylation of protein sequences containing an existing glycan modification is observed for some GalNAc-Ts due to a lectin domain (*Hassan et al., 2000*; *Kubota et al., 2006*; *Revoredo et al., 2016*) and Minerva could theoretically affect such a GalNAc-T in *Drosophila*. Alternatively, Minerva, while in the 'outward open' conformation identified for MFS structures (*Quistgaard et al., 2016*), may itself have a lectin-like interaction with Tn and T glycoforms that have already been added on a loop of particular proteins. Minerva's binding could open up the target protein's conformation to increase or block access to other potential glycosites and thus affect the final glycosylation state on select glycoproteins.

The changes we see in O-glycosylation are also likely due to a combination of Minerva's direct and indirect effects. O-GalNAc modification of vertebrate Notch can affect Notch signaling during development (*Boskovski et al., 2013*); the *Drosophila* ortholog of the responsible GalNAc transferase is also essential for embryogenesis (*Bennett et al., 2010*; *Schwientek et al., 2002*). A GalNAcT in *Xenopus* can glycosylate a peptide corresponding to the ActR IIB receptor and inhibit Activin and BMP type signaling (*Herr et al., 2008*; *Voglmeir et al., 2015*). Thus the changed glycosylation we observe on components of the Notch and Dpp pathways could alter transcription (*Hamaratoglu et al., 2014*; *Ntziachristos et al., 2014*), shifting protein levels and thereby changing the ratio of some glycopeptides in the *mrva* mutant relative to the wild type. Proteins in which glycosylation at other sites is unchanged or changed in the opposite direction are those most likely to be directly affected by Minerva. Such proteins include ones involved in protein folding and O-glycan addition and removal (*Figure 4H*) (*Tien et al., 2008*). If changes in the glycosylation of these proteins alters their specificity or activity, some of the shifts we observe in our glycoproteomic analysis could be indirect in a different way; an initial effect of Minerva on the glycosylation of regulators of protein folding and glycosylation could change how these primary Minerva targets affect the glycosylation of a second wave of proteins.

## An invasion program regulated by Minerva

The truncated immature core1 T and Tn O-glycans are not usually present in normal human tissues but exposure of these uncapped glycans has been found on the majority of cancers and serves as a negative indicator of patient outcome (*Fu et al., 2016*; *Springer, 1984*). Increases in Tn antigen due to a shift in GalNAcT localization to the ER promote invasion and metastasis (*Gill et al., 2013*). An antibody against T antigen has decreased the metastatic spread of cancer cells in mice (*Heimburg et al., 2006*). Here we further strengthen the case for a causative relationship between T antigen modification and the invasive migration that underlies metastasis. The transient appearance of T antigen in human fetuses (*Barr et al., 1989*) and the conserved function of Minerva lead us to propose that the change in O-glycosylation in cancer represents the reactivation of an ancient developmental program for invasion. Our embryonic glycoproteome analysis identifies 106 T antigen modified proteins, a very large set to investigate. However, the absence of Mrva causes invasion defects and deficits in T antigen modification on only 10–20 proteins; these include components involved in protein folding, glycosylation modification, and the signaling pathways triggered by Notch and the BMP family member, Dpp.

Our working model is that the defect in germband tissue invasion seen in the *mrva* mutant is caused by the absence of T antigen on this group of proteins that act coordinately (*Figure 6H*). 56% of these have vertebrate orthologs, and 55% of those have already been linked to cancer and metastasis. The vertebrate ortholog of Qsox1, the protein with the largest changes in T antigen in the *mrva* mutant, can enhance cancer cell invasion in *in vitro* assays and higher levels of the protein have been associated with poor patient outcomes (*Katchman et al., 2013*; *Katchman et al., 2011*). We find that the strongest effect of *Drosophila* Qsox1 on macrophage migration is to reduce the time by two fold that macrophages take sitting at the germband edge before they successfully begin to invade into the germband tissues. We also observe in *qsox1* and *mrva* mutants that LanA levels are higher within the macrophages and somewhat elevated near but not at the macrophage cell edges. This could be due to some combination of the following shifts in cellular processes: an increase in LanA production, a decrease in its degradation, a slowing of its secretion or a speeding of its diffusion. We base our model on the functions that have been previously defined for the Qsox1 sulfhydryl oxidase family, in integrating laminin into the ECM (*Ilani et al., 2013*) and aiding secretion of EGF domains (*Tien et al., 2008*) which are found in *Drosophila* Laminins. If Qsox1 is needed for the efficient secretion and integration of LanA into the ECM, its absence could result in a less robustly cross-linked matrix. ECM crosslinking has been shown to enhance Integrin signaling, focal adhesion formation, and invasion of mammalian tumor cells (*Leventhal et al., 2009*). In its absence *Drosophila* macrophages which utilize Integrin during invasion (*Siekhaus et al., 2010*) and whose invasive migration is accompanied by deformation of the flanking tissue (*Ratheesh et al., 2018*), could be unable to generate sufficient traction forces to enter. Indeed, mutating another subunit of the *Drosophila* Laminin trimer, LanB1, reduces both normal LanA deposition and germband invasion by macrophages (*Matsubayashi et al., 2017*; *Sánchez-Sánchez et al., 2017*). A determination of the effect of Minerva's regulation awaits a characterization of Qsox1 mutated such that it is incapable of being modified by T antigen on the Mrva-dependent sites. Nonetheless, the similarity of the changes in LanA we observe in the *mrva*$^{3102}$ and *qsox1*$^{KG046152}$ mutant supports the conclusion that Mrva dependent T-antigen modification of Qsox1 is necessary for its activity on some substrates. Given that *mrva*$^{3102}$ mutants take even longer than *qsox1*$^{KG04615}$ to enter germband tissue and display much stronger defects thereafter, we propose that T antigen modifications on other proteins are also crucial for tissue entry, and underlie the defect in invasive migration within the germband.

Minerva's vertebrate ortholog, MFSD1, can rescue macrophage migration defects and restores higher T antigen levels. Tagged versions of Minerva's vertebrate ortholog, MFSD1, detected the protein in lysosomes in HeLa and rat liver cells (*Chapel et al., 2013*; *Palmieri et al., 2011*). In four metastasizing mouse tumor cell lines we find MFSD1 mainly in the Golgi, where O-glycosylation is known to occur (*Bennett et al., 2012*). We do not yet know if invasion and metastasis is altered by the absence of MFSD1 but will be testing this in future work. Akin to how kinases add phosphogroups to affect a set of proteins and orchestrate a particular cellular response, we propose that Minerva in *Drosophila* macrophages and its vertebrate ortholog MFSD1 in cancer trigger changes in O-glycosylation that coordinately modulate, activate and inhibit a protein group to affect cellular dissemination and tissue invasion.

# Materials and methods

**Key resources table**

| Designation | Source or reference | Identifiers | Additional information |
|---|---|---|---|
| *mrva* | NA | FlyBase:FBgn0035763 | |
| *qsox1* | NA | FlyBase: FBgn0033814 | |
| *C1GalTA* | NA | FlyBase: FBgn0032078 | |
| *srp-Gal4* | PMID: 15239955 | | |
| *srp-3xmCherry* | PMID: 29321168 | RRID:BDSC_78358 and 78359 | |
| *srp-H2A::3xmCherry* | PMID: 29321168 | RRID:BDSC_78360 and 78361 | |
| *UAS-CG8602* ::FLAG::HA | PMID: 22036573 | | |
| *mrva³¹⁰²* | Bloomington Drosophila Stock Center (BDSC), RRID:SCR_006457 | RRID:BDSC_17262 | |
| *Df(3L)BSC117* | BDSC, RRID:SCR_006457 | RRID:BDSC_8976 | |
| *UAS-mCherry.NLS* | BDSC, RRID:SCR_006457 | RRID:BDSC_38425 | |
| *C1GalTA2.1* | BDSC, RRID:SCR_006457 | RRID:BDSC_28834 | |
| *C1GalTA RNAi 1* | Vienna Drosophila Resource Centre (VDRC), RRID:SCR_013805 | VDRC: 2826 | |
| *C1GalTA RNAi 2* | VDRC, RRID:SCR_013805 | VDRC: 110406 | |
| *CG8602 RNAi* | VDRC, RRID:SCR_013805 | VDRC: 101575 | |
| *qsox1RNAi* | VDRC, RRID:SCR_013805 | VDRC: 108288 | |
| *qsox1 KG04615* | BDSC, RRID:SCR_006457 | RRID:BDSC_13824 | |
| MC-38 | Other | | Gift from Borsig lab, Univ of Zurich (UZH) |
| 4T1 | Other | ATCC Cat# CRL-2539, RRID:CVCL_0125 | Gift from Borsig lab, UZH |
| LLC1 | Other | ATCC Cat# CRL-1642, RRID:CVCL_4358 | Gift from Borsig lab, UZH |
| B16-BL6 | Other | NCI-DTP Cat# B16BL-6, RRID:CVCL_0157 | Gift from Borsig lab, UZH |
| S2R+ | Other | | Gift from Frederico Mauri of the Knoblich lab at IMBA, Vienna |
| *srpHemo-CG8602::3xmCherry* | this paper | | CG8602 amplified from genome cloned into DSPL172 (PMID: 29321168) |
| MT-CG8602 ::FLAG::HA | Drosophila Genomic Resource Center (DGRC), RRID:SCR_002845 | DGRC: FMO06045 | |
| MT-Qsox1 ::FLAG::HA | DGRC, RRID:SCR_002845 | DGRC: FMO06379 | |
| PTS1-GFP | Other | | Gift from Dr. McNew |

*Continued on next page*

Continued

| Designation | Source or reference | Identifiers | Additional information |
|---|---|---|---|
| *MFSD1-eGFP* | this paper | | MFSD1 amplified from dendritic cell cDNA library, inserted into Doxycycline inducible expression vector pInducer20 |
| Anti-GFP clone 2B6 | Other | | Gift from Ogris lab, MFPL Vienna; (1:100) for WB |
| Anti-GFP clone 5G4 | Other | | Gift from Ogris lab, MFPL Vienna; (1:50) for immuno chemistry |
| Anti-T-antigen (mouse monoclonal) | PMID: 23584533 | | (1:5 for immunochemistry; 1:10 for WB) |
| Anti-profilin (mouse monoclonal) | Developmental Studies Hybridoma Bank (DSHB), RRID:SCR_013527 | DSHB Cat# chi 1J, RRID:AB_528439 | (1:50) |
| Anti-GAPDH (rabbit monoclonal) | Abcam, RRID:SCR_012931 | Abcam Cat# ab181603, RRID:AB_2687666 | (1:10000) for WB |
| Anti-GRASP65 (rabbit polyclonal) | Thermo Fisher Scientific, RRID:SCR_008452 | ThermoFischer Cat# PA3-910, RRID:AB_2113207 | (1:200) for immuno chemistry |
| Anti-Rab5 (rabbit monoclonal) | Cell Signaling Technology (CST), RRID:SCR_004431, Clone C8B1 | CST Cat# 3547, RRID:AB_2300649 | (1:200) for immuno chemistry |
| Anti-Rab7 (rabbit monoclonal) | CST, RRID:SCR_004431, Clone D95F2 | CST Cat# 9367, RRID:AB_1904103 | (1:200) for immuno chemistry |
| Anti-LAMP1 (rabbit polyclonal) | Abcam, RRID:SCR_012931 | Abcam Cat# ab24170, RRID:AB_775978 | (1:200) for immuno chemistry |
| Anti- Cnx99a (mouse monoclonal) | DSHB, RRID:SCR_013527 | DSHB Cat# Cnx99A 6-2-1, RRID:AB_2722011 | (1:5) |
| Anti- Hrs 27.4 (mouse monoclonal) | DSHB, RRID:SCR_013527 | DSHB Cat# Hrs 27–4, RRID:AB_2618261 | (1:5) |
| Anti- Golgin 84 (mouse monoclonal) | DSHB, RRID:SCR_013527 | DSHB Cat# Golgin84 12–1, RRID:AB_2722113 | (1:5) |
| Anti Rab7 (mouse monoclonal) | DSHB, RRID:SCR_013527 | DSHB Cat# Rab7, RRID:AB_2722471 | (1:5) |
| Anti-GMAP (goat polyclonal) | DSHB, RRID:SCR_013527 | DSHB Cat# GMAP, RRID:AB_2618259 | (1:50) |
| Anti- Golgin 245 (goat polyclonal) | DSHB, RRID:SCR_013527 | DSHB Cat# Golgin245, RRID:AB_2618260 | (1:50) |
| Anti- HA (rat monoclonal) | Roche, RRID:SCR_001326 | Roche Cat# 3F10, RRID:AB_2314622 | (1:50) |
| Anti-LanA (rabbit polyclonal) | PMID:9257722 | | gift from Stefan Baumgartner (1:500) |
| Anti-Vasa (rat monoclonal) | DSHB, RRID:SCR_013527 | DSHB Cat# anti-vasa, RRID:AB_760351 | (1:25) |
| Alexa 488- or 557- or 633- secondaries | Thermo Fisher Scientific, RRID:SCR_008452 | | (1:500 for 488 and 557; 1:100 for 633) |
| Goat-anti-rabbit IgG (H + L)-HRP | BioRad | Bio-Rad Cat# 170–6515, RRID:AB_2617112 | (1:10000) |

*Continued on next page*

*Continued*

| Designation | Source or reference | Identifiers | Additional information |
|---|---|---|---|
| Goat-anti-mouse IgG (H/L):HRP | BioRad | Bio-Rad Cat# 170–6516, RRID:AB_11125547 | (1:10000) |
| LysoTracker Green DND-26 | Invitrogen, RRID:SCR_008410 | L7526 | 75 nM |
| Alexa Fluor 488 Phalloidin | Invitrogen, RRID:SCR_008410 | A12379 | (1:500) |
| Vectashield mounting medium | Vector Laboratories, RRID:SCR_000821 | VectorLabs: H-1000 | |
| Vectashield mounting medium with DAPI | Vector Laboratories, RRID:SCR_000821 | VectorLabs: H-1200 | |
| Halocarbon Oil 27 | Sigma-Aldrich, RRID:SCR_008988 | Sigma Aldrich: Cat# H8773 | |
| *srpHemo-mrva* | this paper | | CG8602 amplified from genome cloned into srpHemo plasmid |
| *srpHemo-MFSD1* | this paper | | mmMFSD1 amplified from dendritic cell cDNA library cloned into srpHemo plasmid |
| Mrva fw | Fly Primer Bank | | qPCR; 5'TGTGCTTCG TGGGAGGTTTC |
| Mrva rv | Fly Primer Bank | | qPCR; 5'GCAGGCAAA GATCAACTGACC |
| C1GalTA fw | Fly Primer Bank | | qPCR; 5' TGCCAACAGTC TGCTAGGAAG |
| C1GalTA rv | Fly Primer Bank | | qPCR: 5'CTGTGATGT GCATCGTTCACG |
| Ugalt fw | Fly Primer Bank | | qPCR; 5'GCAA GGATG CCCAGAAGTTTG |
| Ugalt rv | Fly Primer Bank | | qPCR; 5'GAT ATAGACC AGCGAGGGGAC |
| RpL32 fw | Fly Primer Bank | | qPCR; 5'AGC ATACAGG CCCAAGATCG |
| RpL32 rv | Fly Primer Bank | | qPCR; 5'TGT TGTCGATA CCCTTGGGC |
| Lectin staining kit #2 | EY Laboratories | EYLabs:FLK-002 | |
| FIJI | http://fiji.sc/ RRID:SCR_002285) | | |
| Imaris | http://www.bitplane.com/imaris/imaris, RRID:SCR_007370 | | |
| Matlab | https://www.mathworks.com/products/matlab.html, RRID:SCR_001622 | | |

*Continued*

| Designation | Source or reference | Identifiers | Additional information |
| --- | --- | --- | --- |
| FlowJo | https://www.flowjo .com/RRID:SCR_008520 | | |
| LaVision ImSpector | http://www.lavision biotec.com/, RRID:SCR_015249 | | |
| Proteome Discoverer 1.4 | https://www.thermo fisher.com/order/ catalog/product/ OPTON-30795, RRID:SCR_014477 | | |
| LightCycler 480 software | https://lifescience.roche .com/en_at/products/ lightcycler14301-480 -software-version-15.html | | |
| GraphPad Prism | https://www.graphpad.com/ scientific-software/prism/ RRID:SCR_002798 | | |

## Fly work

Flies were raised on food bought from IMBA (Vienna, Austria) which contained the standard recipe of agar, cornmeal, and molasses with the addition of 1.5% Nipagin. Adults were placed in cages in a Percival DR36VL incubator maintained at 29°C and 65% humidity; embryos were collected on standard plates prepared in house from apple juice, sugar, agar and Nipagin supplemented with yeast from Lesaffre (Marcq, France) on the plate surface. Embryo collections for fixation (7 hr collection) as well as live imaging (4.5 hr collection) were conducted at 29°C.

### Fly Lines utilized

*srpHemo-GAL4* was provided by K. Brückner (UCSF, USA) (*Bruckner et al., 2004*), *UAS-CG8602::FLAG::HA* (from K. VijayRaghavan National Centre for Biological Sciences, Tata Institute of Fundamental Research) (*Guruharsha et al., 2011*). The stocks $w^{1118}$; *minerva*$^{3102}$ (BDSC-17262), ($pn^1$;; $ry^{503}Dr^1$P[Δ 2–3] (BDSC-1429), *Df(3L)BSC117* (BDSC-8976), *Oregon R* (BDSC-2375), $w^-$; *P{w[+mC] =UAS mCherry.NLS}2;MKRS/Tm6b, Tb(1)* (BDSC-38425), $w^-$,*P{UAS-Rab11-GFP}2* (BDSC-8506), *y(1) sc[\*] v(1); P{y[+t7.7] v[+t1.8]=TRiP .GL00069}attP2* (BDSC-35195), *y(1) w[\*]; Mi{y[+mDint2]=MIC} GlcAT-P[MI05251]/TM3, Sb(1)* (BDSC-40779) were obtained from the Bloomington *Drosophila* Stock Centre, Bloomington, USA. The RNAi lines v60100, v110406, v2826, v101575 were obtained from the Vienna Drosophila Resource Center (VDRC), Vienna, Austria. Lines $w^-$; *P{w[+mC]; srpHemo-3xmCherry}*, $w^-$; *P{w[+mC]; srpHemo-H2A::3xmCherry}* were published previously (*Gyoergy et al., 2018*).

## Exact genotype of *Drosophila* lines used in figures

*Figure 1D-H: w-; +; srpHemo-3xmCherry. Figure 1I-K*: Control: *w- P(w+)UAS-dicer/w-; P{attP,y[+],w[3']/+; srpHemo-Gal4 UAS-GFP UAS-H2A::RFP/+.* C1GalTA RNAi: *w⁻ P(w+)UAS-dicer2/w-; RNAi C1GalTA (v110406)/+; srpHemo-Gal4 UAS-GFP UAS-H2A:RFP/+. Figure 1L*: Control: *w-; +; srpHemo-H2A::3xmCherry.* C1GalLTA mutant: *w-; C1GalTA$^{2.1}$; srpHemo-H2A::3xmCherry. Figure 1M*: Control: *w-; srpHemo-H2A::3xmCherry.* GlcAT-P mutant: *w-; srpHemo-H2A::3xmCherry, Mi{MIC}GlcAT-PMI05251.*

*Figure 1—figure supplement 1A–L: w-; +; srpHemo-3xmCherry. Figure 1—figure supplement 1M, N, P*: Control: *w- UAS-Dicer2/w-; P{attP,y[+]w[3']/+; srpHemo-Gal4 UAS-GFP UAS-H2A::RFP/+.* C1GalTA RNAi: *w-UAS-Dicer2/w-; RNAi C1GalTA (v110406)/+; srpHemo-Gal4 UAS-GFP UAS-H2A:: RFP/+. Figure 1—figure supplement 1O*: Control: *w-; +; srpHemo-H2A::3xmCherry.* C1GalTA mutant: *w-; C1GalTA$^{2.1}$; srpHemo-H2A::3xmCherry. Figure 1—figure supplement 1P*: Control: *w- UAS-Dicer2/w-; P{attP,y[+]w[3']/+; srpHemo-Gal4 UAS-GFP UAS-H2A::RFP/+.* C1GalTA RNAi: *w- UAS-Dicer2/w-; RNAi C1GalTA (v2826)/+; srpHemo-Gal4 UAS-GFP UAS-H2A::RFP/+. Figure 1—*

*figure supplement 1Q*: Control: *w-; srpHemo-H2A::3xmCherry*. GlcAT-P mutant: *w-; srpHemo-H2A::3xmCherry; Mi{MIC}GlcAT-PMI05251*.

*Figure 2A, B, D*: *w-; +; srpHemo-3xmCherry*. *Figure 2E, F, G*: Control: *w-; +; srpHemo-3xmCherry*. CG8602 mutant: *w-; +; srpHemo-3xmCherry,P{EP}CG8602³¹⁰²*. *Figure 2I*: *w-; srpHemo-Gal4; UAS-CG8602::FLAG::HA*.

*Figure 3A*: *w-; +; srpHemo-H2A::3xmCherry*. *Figure 3B*: *w-; +; srpHemo-H2A::3xmCherry, P{EP}CG8602³¹⁰²*. *Figure 3C*: *w-; srpHemo-CG8602; srpHemo-H2A::3xmCherry P{EP}CG8602³¹⁰²*. *Figure 3D*: Control: *w-; srpHemo-Gal4 UAS-mCherry::nls; +*. CG8602 (Mrva) mutant: *w-; srpHemo-Gal4 UAS-mCherry::nls; P{EP}CG8602³¹⁰²*, Df cross: *w-; srpHemo-Gal4 UAS-mCherry:nls; P{EP}CG8602³¹⁰²/Df(3L)BSC117*. Rescue: *w-; srpHemo-Gal4 UAS-mCherry:nls; UAS-CG8602::FLAG::HA P{EP}CG8602³¹⁰²*. Precise excision: *srpHemo-Gal4 UAS-mCherry:nls; P{EP}CG8602³¹⁰²Δ32*. *Figure 3E*: Control: *w⁻ P(w+)UAS-dicer/+; +; srpHemo-Gal4 UAS-GFP UAS-H2A:RFP/+*. Mrva RNAi: *w- UAS-dicer2/w-; RNAi CG8602 (v101575)/+; srpHemo-Gal4 UAS-GFP UAS-H2A:RFP/+*. *Figure 3F*: Control: *w-; srpHemo-Gal4 UAS-mCherry::nls; +*. Mrva mutant: *w-; srpHemo-Gal4 UAS-mCherry::nls; P{EP}CG8602³¹⁰²*. *Figure 3G*: Control: *w-; +; srpHemo-3xmCherry*. Mrva mutant: *w-; +; srpHemo-3xmCherry P{EP}CG8602³¹⁰²*. Cadherin Mrva double mutant: *w-; shg^P34; srpHemo-3xmCherry P{EP}CG8602³¹⁰²*. *Figure 3H*: Control: *w-; +; srpHemo-3xmCherry*. Mrva mutant: *w-; +; srpHemo-3xmCherry P{EP}CG8602³¹⁰²*. *Figure 3I-M*: Control: *w-; +; srpHemo-H2A::3xmCherry*. Mrva mutant: *w-; +; srpHemo-H2A::3xmCherry P{EP}CG8602³¹⁰²*.

*Figure 3-figure supplement 1A:* Control: *w-; +; srpHemo-H2A::3xmCherry*. Mrva mutant: *w-; +; srpHemo-H2A::3xmCherry P{EP}CG8602³¹⁰²*. Rescue: *w-; srp-CG8602; srpHemo-H2A::3xmCherry P{EP}CG8602³¹⁰²*. *Figure 3-figure supplement 1B, C, E*: Control: *w-; +; srpHemo-Gal4 UAS-GFP UAS-H2A:RFP/+*. Mrva RNAi: *w-; RNAi CG8602 (v101575)/+; srpHemo-Gal4 UAS-GFP UAS-H2A::RFP/+*. *Figure 3-figure supplement 1D, F-G*: Control: *w-; +; srpHemo-H2A::3xmCherry*. Mrva mutant: *w-; +; srpHemo-H2A::3xmCherry P{EP}CG8602³¹⁰²*. *Figure 3-figure supplement 1H-L*: Control: *w-; +; srpHemo-3xmCherry*. Mrva mutant: *w-; +; srpHemo-3xmCherry P{EP}CG8602³¹⁰²*

*Figure 4A-I*: Control: *w-; +, srpHemo-3xmCherry*. Mrva mutant: *w-; +, srpHemo-3xmCherry P{EP}CG8602³¹⁰²*.

*Figure 5A-B:* Control: *w-; +; srpHemo-3xmCherry*. Qsox1 mutant: *w-;P{SUPor-P}Qsox1KG04615; srpHemo-3xmCherry*. *Figure 5C*: *w/y,w[1118]; P{attP,y[+],w[3']}/srpHemo-Gal4; srpHemo-H2A::3xmCherry/+*.Qsox1 RNAi: *w-/y,w[1118]; v108288/srpHemo-Gal4; srpHemo-H2A::3xmCherry/+*. *Figure 5D-G*: Control: *w-; +; srpHemo-H2A::3xmCherry*. Qsox1 mutant: *w-;P{SUPor-P}Qsox1KG04615; srpHemo-H2A::3xmCherry*. *Figure 5J*: Control: *w-; +; srpHemo-3xmCherry*. Mrva mutant: *w-; +; srpHemo-3xmCherry P{EP}CG8602³¹⁰²*. Qsox1 mutant: *w-; P{SUPor-P}Qsox1KG04615;srpHemo-3xmCherry*.

*Figure 5-figure supplement 1A-B*: Control: *w-; +; srpHemo-3xmCherry*. Qsox1 mutant: *w-;P{SUPor-P}Qsox1KG04615; srpHemo-3xmCherry*. *Figure 5-figure supplement 1C, F*: *w-; +; srpHemo-H2A::3xmCherry, w-; P{SUPor-P}Qsox1KG04615; srpHemo-H2A::3xmCherry*. *Figure 5-figure supplement 1D-E*: Control: *w-/y,w[1118]; P{attP,y[+],w[3']}/srpHemo-Gal4; srpHemo-H2A::3xmCherry/+*. Qsox1 RNAi: *w-/y,w[1118]; v108288/srpHemo-Gal4; srpHemo-H2A::3xmCherry/+*. Figure 5-figure supplement 1K-N: Control: *w-; +; srpHemo-3xmCherry*. Mrva mutant: *w-; +; srpHemo-3xmCherry P{EP}CG8602³¹⁰²*, Qsox1 mutant: *w-; P{SUPor-P}Qsox1KG04615; srpHemo-3xmCherry*.

*Figure 6D*: *w-; srpHemo-MFSD1; srpHemo-H2A::3xmCherry P{EP}CG8602³¹⁰²*. *Figure 6E*: Control: *w-; +; srpHemo-H2A::3xmCherry*. Mrva mutant: *w-; +; srpHemo-H2A::3xmCherry P{EP}CG8602³¹⁰²*. MFSD1 rescue: *w-; srpHemo-MFSD1; srpHemo-H2A::3xmCherry P{EP}CG8602³¹⁰²*. *Figure 6F, G*: Control: *w-; +; srpHemo-3xmCherry*. Mrva mutant: *w-; +; srpHemo-3xmCherry P{EP}CG8602³¹⁰²*. MFSD1 rescue: *w-; srpHemo-MFSD1; srpHemo-3xmCherry P{EP}CG8602³¹⁰²*.

## Embryo fixation and immunohistochemistry

Embryos were collected on apple juice plates from between 6 and 8.5 hr at 29°C. Embryos were incubated in 50% Chlorox (DanClorix) for 5 min and washed. Embryos were fixed with 17% formaldehyde/heptane for 20 min followed by methanol or ethanol devitellinization except for T antigen analysis, when embryos were fixed in 4% paraformaldehyde/heptane. Fixed embryos were blocked in BBT (0.1M PBS + 0,1% TritonX-100 +0,1% BSA) for 2 hr at RT. Antibodies were used at the following dilutions: α-T antigen (*Steentoft et al., 2011*) 1:5, α-GFP (Aves Labs Inc., Tigard, Oregon) 1:500; α-

LanA (*Kumagai et al., 1997*) (a gift from Stefan Baumgartner) 1:500; α-Vasa (*Aruna et al., 2009*) (DSHB, deposited by A. Sprading/D. Williams) 1:25; and incubated overnight at 4°C (GFP) or room temperature (T antigen, LanA). Afterwards, embryos were washed in BBT for 2 hr, incubated with secondary antibodies (Thermo Fisher Scientific, Waltham, Massachusetts, USA) at RT for 2 hr, and washed again for 2 hr. Vectashield (Vector Laboratories, Burlingame, USA) was then added. After overnight incubation in Vectashield at 4°C, embryos were mounted on a slide and imaged with a Zeiss Inverted LSM700 Confocal Microscope using a Plan-Apochromat 20X/0.8 Air Objective or a Plan-Apochromat 63X/1.4 Oil Objective.

## Ovary dissection and immunostaining

3–5 day old females were fed with yeast for 2 days at 25°C. For ovary dissection, females were anesthetized using the FlyNap anesthetic kit (Carolina, Burlington, NC, USA) and further transferred to ice cold PBS in which ovaries were extracted with pre-cleaned forceps. Individual ovaries were fixed in 4% Paraformaldehyde/PBS at room temperature (RT) for 20 min with agitation. Three wash steps with PBS at RT for 10 min were performed and individual ovaries were incubated in PBS supplemented with 0.1% of Triton X-100 (PBT) for 10 min at RT to allow permeabilization of the tissue. Ovaries were incubated in phalloidin-A488 (Thermo Fisher) diluted in PBT (1:300) overnight at 4°C. After being washed with PBT and PBS, ovaries were mounted in Vectashield + DAPI (LifeTechnologies, Carlsbad, USA).

## Fixed ovary image analysis for border cell migration

Ovaries were imaged as a Z-series (1 µm apart) with a Plan-Apochromat 20X/0.8 Air Objective on a Zeiss LSM700 inverted microscope. Images were acquired from stage 10 oocytes and maximum-intensity projections were created using ImageJ (NHI, USA). Border cells were identified by the clustered nuclei and their enriched actin staining. Border cell migration was quantified in the DAPI images as the percentage observed relative to the expected migration to the edge of the oocyte for these cells in stage 10 oocytes. Measurements were performed using ImageJ software (NIH, USA).

## Lectin staining

Embryos were fixed with 10% formaldehyde/heptane and devitellinized with Ethanol. Blocking was conducted in BBT for 2 hr at room temperature. A FITC-labeled lectin kit #2 (EY laboratories, San Mateo, CA, USA) was utilized (table below summarizes abbreviations of used lectins). Each lectin was diluted to 1:25 and incubated with fixed embryos overnight at room temperature (RT). Embryos were washed in BBT for 2 hr at RT and Vectashield was added. After overnight incubation at 4°C, embryos were mounted on a slide and imaged with a Zeiss Inverted LSM700 Confocal Microscope using a Plan-Apochromat 63X/1.4 Oil Objective. Macrophages in late Stage 11 embryos were imaged at germband entry and evaluated by eye for enriched staining on macrophages compared to other tissues.

| Lectin | Peanut agglutinin | Ulex europaeus agglutinin | Wheat germ agglutinin | Griffonia simplicifolia agglutinin I | Maclura pomifera agglutinin | Griffonia simplicifolia agglutinin II |
|---|---|---|---|---|---|---|
| Abbreviation | PNA | UEA-I | WGA | GS-I | MPA | GS-II |
| Lectin | Soybean agglutinin | Dolichos biflorus agglutinin | Concanavalin A | Helix pomatia agglutinin | Limulus poly-phenus agglutinin | Bauhinia purpurea agglutinin |
| Abbreviation | SBA | DBA | ConA | HPA | LPA | BPA |

## Macrophage extraction

Embryos were bleached in 50% Chlorox in water for 5 min at RT. Stage late 11/early 12 embryos were lined up and then glued to 50 mm Dish No. 0 Coverslip, 14 mm Glass Diameter, Uncoated dish (Zeiss, Germany). Cells from the germband margin were extracted using a ES Blastocyte Injection Pipet (spiked, 20 µm inner diameter, 55 mm length; BioMedical Instruments, Germany). Extracted cells were placed in Schneider's medium (Gibco, Dublin, Ireland) supplemented with 20% FBS (Sigma-Aldrich, Saint Louis, Missouri, USA).

## Immunohistochemistry of extracted macrophages

Extracted macrophages were collected by centrifugation at 500 g for 5 min at room temperature. The cell pellet was resuspended in a small volume of Phospho-buffered saline (PBS) and smeared on a cover slip. The cell suspension was left to dry before cells were fixed with 4% paraformaldehyde in 0.1M Phosphate Buffer for 20 min at room temperature. Cells were washed 3 times in 0.1M PBS and permeabilized in 0.5% Triton-X 100 in PBS. Cells were blocked for 1 hr at room temperature in 20% Fetal Bovine Serum +0.25% Triton X-100 in PBS. Primary antibodies were diluted in blocking buffer: anti-HA (Roche, Basel, Switzerland) 1:50, anti-Golgin 84, 1:25, anti-Calnexin 99a 1:25, anti-Hrs.8.2 1:25 or anti-Rab7 1:25 all from DSHB (*Riedel et al., 2016*), and incubated for 1 hr at room temperature. Cells were then washed 5 times in blocking buffer. Secondary antibodies were diluted in blocking buffer: anti-rat 633 1:300, anti-mouse 488 1:300 (both from ThermoFisher Scientific, Waltham, Massachusetts, USA). Secondary antibodies were incubated for 1 hr at room temperature. Cells were washed 5 times in PBS + 0.1% Triton X-100 and mounted in VectaShield + DAPI (LifeTechnologies, Carlsbad, USA) utilized at 1:75.

## S2 cell work

S2R+ cells (a gift from Frederico Mauri of the Knoblich laboratory at IMBA, Vienna) were grown in Schneider's medium (Gibco) supplemented with 10% FBS (Gibco) and transfected with PTS1-GFP (a gift from Dr. McNew) and/or the *srpHemo-CG8602::3xmCherry* construct using Effectene Tranfection Reagent (Qiagen, Hilden, Germany) following the manufacturer's protocol. Transfected S2R + cells were grown on Poly-L-Lysine coated coverslips (ThermoFisher Scientific, Waltham, Massachusetts, USA) in complete Schneider's medium (Gibco) supplemented with 10% FBS (Sigma-Aldrich, Saint Louis, Missouri, USA) and 1% Pen/Strep (Gibco) to a confluency of 60%. To visualize lysosomes, cells were incubated with Lysotracker 75 nM Green DND-26 (Invitrogen) in complete Schneider's medium for 30 min at 25°C. Cells were washed in complete Schneider's medium 3 times before imaging on an inverted LSM-700 (Zeiss). To visualize mitochondria, mitotracker Green FM (Invitrogen, Carlsbad, CA, USA) was diluted in prewarmed Schneider's medium supplemented with 1% Pen/Strep to a concentration of 250 nM. Cells were incubated in the Mitotracker solution for 45 min at 25°C. Cells were then washed 3 times in complete Schneider's medium before imaging.

To visualize Golgi, ER, early and late endosomes as well as the nucleus, S2R+ cells were transfected with MT-CG8602::FLAG::HA (DGRC: FMO06045) or MT-Qsox1::FLAG::HA (DGRC: FMO06379) with Effectene Tranfection Reagent (Qiagen) following the manufacturer's protocol. 24 hr after transfection gene expression was induced by addition of 1 mM Cu2SO4 (Sigma) and cells were incubated for an additional 24 hr. Cells were then fixed in 4% PFA (Sigma) in 0.1M PB for 20 min at room temperature, permeabilized in 0.5% Triton X-100 (Sigma) in PBS for 15 min and blocked for 2 hr in 20% FBS (Sigma), 0.25% Triton X-100 in PBS at room temperature.

Cells were then stained with anti-HA antibody 1:50 (Roche) and either anti-Cnx99a (1:5), anti-Hrs 8.2 (1:5), anti-Golgin 84 (1:5), anti-Rab7 (1:5), anti- GMAP (1:50) or anti- Golgin 245 (1:50) (all antibodies from DSHB) (*Riedel et al., 2016*). Cells were washed in 20% FBS (Sigma), 0.25% Triton X-100 in PBS 5 times and then incubated with anti-rat Alexa Fluor 633 1:50 and either anti-mouse Alexa Fluor 488 or anti-goat Alexa Fluor 488 1:100 (Thermo Fisher) for 2 hr at room temperature. Cells were washed again 5 times and then mounted in Vectashield Mounting Medium +DAPI (Vector Laboratories) and imaged with Zeiss LSM 700 or 800 confocal microscopes. Quantitation of colocalization was performed as indicated below.

The cell line was routinely tested for Mycoplasm infection and found to be negative.

## DNA isolation from single flies

Single male flies were frozen for at least 3 hr before grinding them in 100 mM Tris-HCl, 100 mM EDTA, 100 mM NaCL and 0.5% SDS. Lysates were incubated at 65°C for 30 min. Then 5M KAc and 6M LiCl were added at a ratio of 1:2.5 and lysates were incubated on ice for 10 min. Lysates were centrifuged for 15 min at 20,000xg, supernatant was isolated and mixed with Isopropanol. Lysates were centrifuged again for 15 min at 20.000xg, supernatant was discarded and the DNA pellet was washed in 70% EtOH and subsequently dissolved in ddH20.

## FACS sorting

Embryos were collected for 1 hr and aged for an additional 5 hr, all at 29°C. Embryos collected from w- flies were processed in parallel and served as a negative control. Embryos were dissociated as described previously (*Gyoergy et al., 2018*). The cells were sorted using a FACS Aria III (BD) flow cytometer. Emission filters were 600LP, 610/20 and 502 LP, 510/50. Data were analyzed with FlowJo software (Tree Star). The cells from the dissociated negative control $w^-$ embryos were sorted to set a baseline plot.

## qPCR

RNA was isolated from approximately 50,000 mCherry positive or mCherry negative FACS sorted macrophages using RNeasy Plus Micro Kit (Qiagen, Hilden, Germany) following manufacturer's protocol. RNA was also isolated from 50 to 100 mg of ovaries (about 15–20 pairs of ovaries extracted as indicated above). Ovaries were homogenized with a pellet homogenizer (VWR, Radnor, USA) and plastic pestles (VWR, Radnor, USA) in 1 ml of Trizol (Thermo Fisher Scientific, Waltham, MA, USA) and centrifuged at 12,000xg for 5 min at 4°C. Further steps were according to the manufacturers protocol. The resulting RNA was used for cDNA synthesis using Sensiscript RT Kit (macrophages) or Omniscript (ovaries) (Qiagen, Hilden, Germany) and oligo dT primers. A Takyon qPCR Kit (Eurogentec, Liege, Belgium) was used to mix qPCR reactions based on the provided protocol. qPCR was run on a LightCycler 480 (Roche, Basel, Switzerland) and data were analyzed in the LightCycler 480 Software and Prism (GraphPad Software). Data are represented as relative expression to a housekeeping gene ($2^{-\Delta ct}$) or fold change in expression ($2^{-\Delta\Delta ct}$). Primer sequences utilized for flies were obtained from the FlyPrimerBank (http://www.flyrnai.org/FlyPrimerBank). Minerva/CG8602: Fw pr TGTGC TTCGTGGGAGGTTTC, Rv pr GCAGGCAAAGATCAACTGACC. C1GalTA: Fw pr TGCCAACAGTC TGCTAGGAAG, Rv pr CTGTGATGTGCATCGTTCACG. Ugalt: Fw pr GCAAGGATGCCCAGAAG TTTG, Rv pr GATATAGACCAGCGAGGGGAC. RpL32: Fw pr AGCATACAGGCCCAAGATCG, Rv pr TGTTGTCGATACCCTTGGGC

## Protein preps from embryos for western blots

Embryos were collected for 7 hr at 29°C, bleached and hand-picked for the correct stage. 50–200 embryos were smashed in RIPA buffer (150 mM NaCl, 0,5% Sodiumdeoxychalat, 0,1% SDS, 50 mM Tris, pH 8) with Protease inhibitor (Complete Mini, EDTA free, Roche, Basel, Switzerland) using a pellet homogenizer (VWR, Radnor, USA) and plastic pestles (VWR, Radnor, USA) and incubated on ice for 30 min. Afterwards, samples were centrifuged at 4°C, 16,000 g for 30 min and the supernatant was collected and used for experiments. The protein concentration was quantified using the Pierce BCA Protein Assay Kit (ThermoFisher Scientific).

## Western blots

30 µg of protein samples were loaded on a 4–15% Mini-PROTEAN TGX Precast Protein Gel (Bio-Rad, Hercules, USA) and run at 100V for 80 min in 1x running buffer (25 mM Tris Base, 190 mM glycine and 0.1%SDS) followed by transfer onto Amersham Protran Premium 0.45 µm NC (GE Healthcare Lifescience, Little Chalfont, UK) or Amersham Hybond Low Fluorescence 0.2 µm PVDF (GE Healthcare Lifescience, Little Chalfont, UK) membrane using a wet transfer protocol with 25 mM Tris Base, 190 mM Glycine +20% MeOH at either 100 Volts for 60 min or 200mA for 90 min at Mini Trans-Blot Cell Module (Bio-Rad, Hercules, USA). Membranes were blocked in PBS-T (0.1% Triton X-100 in PBS) containing 2% BSA or Pierce Clear Milk Blocking Buffer (ThermoFisher Scientific) for 1 hr at RT. Primary antibodies were incubated overnight at 4°C at the following concentrations: α-T antigen (Copenhagen) 1:10, α-profilin (*Verheyen and Cooley, 1994*), DSHB) 1:50, anti-GFP (clone 2B6, Ogris lab, MFPL), anti-GAPDH (ab181603, Abcam, Cambridge, UK). Afterwards, blots were washed 3x for 5 min in blocking solution and incubated with Goat anti Mouse IgG (H/L):HRP (Bio-Rad, Hercules, USA) or goat-anti-rabbit IgG (H + L)-HRP (Bio-Rad, Hercules, USA) at 1:5 000–10,000 for 1–2 hr at room temperature. Blots were washed 2 × 5 min in blocking solution and 1 × 5 min with PBS-T. Blots were developed using SuperSignal West Femto Maximum Sensitivity Substrate (ThermoFisher Scientific, Waltham, Massachusetts, USA) according to manufacturer's instructions. Chemiluminescent signal was detected using the Amersham Imager 600 (GE Healthcare Lifescience) or VersaDoc (Bio-Rad). Images were processed with ImageJ.

## Western blot analysis of S2R+ supernatant

S2R+ cells were transfected as described previously with srpGal4 UAS-Qsox1::FLAG::HA. 2 days post-transfection, medium was removed and cells were washed with PBS. Afterwards, serum-free S2 medium was added and incubated for approximately 40 hr. Afterwards, supernatant was collected and concentrated using Amicon Ultra-4 10K Centrifugal Filter Device (Merck, Kenilworth, New Jersey, United States) to gain 80 µl of concentrated supernatant. 20 µl of supernatant was loaded on gel and analyzed by anti-HA (1:200, Roche). Images were processed with ImageJ.

## Time-lapse imaging, tracking, speed, persistence and germband entry analysis

Embryos were dechorionated in 50% bleach for 5 min, washed with water, and mounted in halocarbon oil 27 (Sigma-Aldrich, Saint Louis, Missouri, USA) between a coverslip and an oxygen permeable membrane (YSI). The anterior dorsolateral region of the embryo was imaged on an inverted multiphoton microscope (TrimScope II, LaVision) equipped with a W Plan-Apochromat 40X/1.4 oil immersion objective (Olympus). mCherry was imaged at 1100 nm excitation wavelengths, using a Ti-Sapphire femtosecond laser system (Coherent Chameleon Ultra) combined with optical parametric oscillator technology (Coherent Chameleon Compact OPO). Excitation intensity profiles were adjusted to tissue penetration depth and Z-sectioning for imaging was set at 1 µm for tracking and segmentation respectively. For long-term imaging, movies were acquired for 132–277 min with a frame rate of 40 s. All embryos were imaged with a temperature control unit set to 28.5°C.

Images acquired from multiphoton microscopy were initially processed with InSpector software (LaVision Bio Tec) to compile channels from the imaging data, and the exported files were further processed using Imaris software (Bitplane) to visualize the recorded channels in 3D. Macrophage speed and persistence were calculated by using embryos in which the macrophage nuclei were labeled with *srpHemo-H2A::3XmCherry* (Gyoergy et al., 2018). The movie from each imaged embryo was rotated and aligned along the AP axis for tracking analysis. Increasing the gain allowed determination of germband position from the autofluorescence of the yolk. Movies for vnc analysis were analyzed for 2 hr from the time point that cells started to dive into the channels to reach the outer vnc. Macrophage nuclei were extracted using the spot detection function and nuclei positions in xyz-dimensions were determined for each time point and used for further quantitative analysis. Cell speeds and directionalities were calculated in Matlab (The MathWorks Inc., Natick, Massachusetts, USA) from single cell positions in 3D for each time frame measured in Imaris (Bitplane). Instantaneous velocities from single cell trajectories were averaged to obtain a mean instantaneous velocity value over the course of measurement. To calculate directionality values, single cell trajectories were split into segments of equal length (10 frames) and calculated via a sliding window as the ratio of the distance between the macrophage start-to-end location over the entire summed distance covered by the macrophage between successive frames in a segment. Calculated directionality values were averaged over all segments in a single trajectory and all trajectories were averaged to obtain a mean directionality value for the duration of measurement, with 0 being the lowest and one the maximum directionality. To estimate the time for entry into the germband, we increased the gain to visualize the germband position from the autofluorescence of the yolk. We assessed the time point when the first macrophage nucleus reached the edge of the germband (taken as T0) and the time point when the first cell nucleus was just within the germband (taken as T1). T1-T0 was defined as the time for macrophage entry.

## Fixed embryo image analysis for T antigen levels

Embryos were imaged with Plan-Apochromat 63X/1.4 Oil Objective on a Zeiss LSM700 inverted. 10 µm stacks (0.5 µm intervals) were taken for properly staged and oriented embryos, starting 10 µm deep in the tissue. These images were converted into Z-stacks in Fiji. ROIs were drawn around macrophages (signal), copied to tissue close by without macrophages (background) and the average intensity in the green channel of each ROI was measured. For each pair of ROIs background was subtracted from signal individually. The average signal from control ROIs from one imaging day and staining was calculated and all data points from control, mutant and rescue from the same set was divided by this value. This way we introduced an artificial value called Arbitrary Unit (AU) that makes it possible to compare all the data with each other, even if they come from different imaging days

when the imaging laser may have a different strength or from different sets of staining. Analysis was done on anonymized samples.

## Macrophage cell counting

Transmitted light images of the embryos were used to measure the position of the germband to determine the stages for analysis. The extent of germband retraction away from the anterior along with the presence of segmentation was used to classify embryos. Embryos with germband retraction of between 29–31% were assigned to late Stage 11. Those with 29–41% retraction (early Stage 12) were analyzed for the number of macrophages that had entered the germband and those with 50–75% retraction (late Stage 12) for the number along the ventral nerve cord (vnc), and in the whole embryo. Macrophages were visualized using confocal microscopy with a Z-resolution of 3 μm and the number of macrophages within the germband or the segments of vnc was calculated in individual slices (and then aggregated) using the Cell Counter plugin in FIJI.

To check that this staging allows embryos from the control and $mrva^{3102}$ mutant to be from the same time during development, embryos were collected for 30 min and then imaged for a further 10 hr using a Nikon-Eclipse Wide field microscope with a Plan-Apochromat 20X/0.5 DIC water Immersion Objective. Bright field images were taken every 5 min, and the timing of the start of the movies was aligned based on when cellularization occurred. We found no significant difference in when germband retraction begins (269.6 ± 9 min in control and 267.1 ± 3 min in $mrva^{3102}$, p=0.75) or in when the germband retracts to 41% (300 ± 9 min for control, 311 ± 5 min in $mrva^{3102}$, p=0.23), or in when the germband retraction is complete (386.5 ± 10 min for control, 401.6 ± 8 min for $mrva^{3102}$, p=0.75). n = 10 embryos for control and 25 embryos for $mrva^{3102}$.

## Cloning

Standard molecular biology methods were used and all constructs were sequenced by Eurofins before injection into flies. Restriction enzymes *BSi*WI, and *Asc*I were obtained from New England Biolabs, Ipswich, Massasuchetts, USA (Frankfurt, Germany). PCR amplifications were performed with GoTaq G2 DNA polymerase (Promega, Madison, USA) using a peqSTAR 2X PCR machine from PEQ-LAB, (Erlangen, Germany). All Infusion cloning was conducted using an Infusion HD Cloning kit obtained from Clontech's European distributor (see above); relevant oligos were chosen using the Infusion primer Tool at the Clontech website.

### Construction of *srpHemo-minerva*

A 1467 bp fragment containing the Minerva (CG8602) ORF was amplified from the UAS-CG8602: FLAG:HA construct (DGRC) using primers Fw GAAGCTTCTGCAAGGATGGCGCGCGAGGACGAG-GAAC, Rv CGGTGCCTAGGCGCGCTATTCAAAGTTCTGATAATTCTCG. The fragment was cloned into the srpHemo plasmid (a gift from Katja Brückner, (*Bruckner et al., 2004*)) after its linearization with AscI, using an Infusion HD cloning kit.

### Construction of *srpHemo-MFSD1*

A 1765 bp fragment containing the MFSD1 ORF was amplified from cDNA prepared from dendritic cells (a gift from M. Sixt's lab) with Fw primer TAGAAGCTTCTGCAACTTTGCTTCCTGCTCCGTTC, Rv primer ATGTGCCTAGGCGCGCGAAGGAAAGGCTTCATCCGCA). The fragment was cloned into the srpHemo plasmid (a gift from Katja Brückner, (*Bruckner et al., 2004*) using an Infusion HD cloning kit (Clontech) after its linearization with AscI (NEB).

### Construction of *srpHemo-mrva::3xmCherry*

Minerva (CG8602) was amplified from a DNA prep from Oregon R flies (Fw primer: AGAGAAGC TTCGTACGCGACAACCCTGCTCTACAGAG; Rv primer CGACCTGCAGCGTACGACCCGATCC TTCAAAGTTCTG). The vector, PCasper4 containing a 3xmCherry construct under the control of the srpHemo promoter (*Gyoergy et al., 2018*), was digested with BsiWI according to the manufacturer's protocol. The vector and insert were homologously recombined using the In-Fusion HD Cloning Kit.

## Generation of pInducer20-MFSD1-eGFP constructs

For C-terminal tagging MFSD1 was PCR amplified from cDNA prepared from dendritic cells (a gift from M. Sixt lab) (Fw primer:GATCTCGAGATGGAGGACGAGGATG; Rv primer: CGACCGGTAAC TCTGGATGAGAGAGC) and digested with XhoI and AgeI (both New England Biolabs, Ipswich, Massasuchetts, USA). This MFSD1 fragment was cloned into XhoI/AgeI digested peGFP-N1 (Addgene, Cambridge, Massachusetts, USA). C-terminally eGFP tagged MFSD1 was further PCR amplified (Fw primer: GGGGACAAGTTTGTACAAAAAAGCAGGCTTAATGGAGGACGAGGAT; Rv primer: GGGGACCACTTTGTACAAGAAAGCTGGGTATTACTTGTACAGCTC). This fragment was cloned using Gateway BP Clonase II Enzyme mix and Gateway LR Clonase II Enzyme Mix (ThermoFisher Scientific, Waltham, Massachusetts, USA) via donor vector pDonR211 into the final Doxycyclin inducible expression vector pInducer20 (*Meerbrey et al., 2011*) according to the manufacturer's instructions. pInducer20-MFSD1-eGFP was amplified in stbl3 bacteria (ThermoFisher Scientific, Waltham, Massachusetts, USA).

## Precise excision

*mrva*$^{3102}$ flies which contain the 3102 P element insert in the 5' region of CG8602 were crossed to a line expressing transposase (BL-1429: *pn*$^1$; *ry*$^{503}$*Dr*$^1$*P[Δ 2–3]*). To allow excision of the P Element, males from the F1 generation containing both the P element and the transposase, were crossed to virgins with the genotype Sp/Cyo; PrDr/TM3Ser (gift from Lehmann lab). In the F2 generation white eyed males were picked and singly crossed to Sp/Cyo; PrDr/TM3Ser virgins.

## LanA quantification

Images were taken with a Z-resolution of 0.5 μm from the head of late stage 12 embryos using a Zeiss LSM800 confocal microscope and a 40x/1.4 Oil DIC objective. A 4 μm long line was drawn over a macrophage with the middle of the line located approximately at the edge of the cell. mCherry and LanA (488) intensities were measured using the Multichannel Plot Profile Plugin in Fiji. Three lines were drawn on each cell to catch the variability of secretion. Only cells standing alone or in small groups that had at least some small visible amount of extracellular LanA were analyzed. From each embryo, 20 cells were analyzed. Images were anonymized before quantification.

## Mammalian cell culture

MC-38 colon carcinoma cells, 4T1 breast carcinoma (ATCC, CRL-2539), Lewis Lung carcinoma LLC1 (ATCC, CRL-1642) and B16-BL6 melanoma (NCI-DTP; B16BL-6) (all gifts from the Borsig lab) were kept in DMEM supplemented with 10% FCS (Sigma-Aldrich, Saint Louis, Missouri, USA), Non-essential Amino Acids, and Na-Pyruvate (Thermo Fisher Scientific, Waltham, Massachusetts, USA). All cells were kept in a humidified incubator at 37°C with 5% CO2. Cells were infected with lentiviral particles containing pInducer20-MFSD1-eGFP. Expression of MFSD1-eGFP was induced with 20 ng/ml (for MC-38) and 100 ng/ml (for 4T1, LLC, B16-BL6) of Doxycycline for 24 hr prior subsequent analysis. Cell lines were routinely tested for Mycoplasm infection and found to be negative. The identity of the cell lines was confirmed by STR analysis by the cell bank from which they were obtained.

## Mammalian cell lysis

Cells were lysed in alysis buffer (25 mM Tris, 150 mM NaCl, 1 mM EDTA, 1% Triton X-100) supplemented with a protease inhibitor cocktail (Complete, Roche, Basel, Switzerland) for 20 min on ice, followed by centrifugation at 14,000x g, 4°C for 5 min. The protein lysates were stored at −80°C. Protein concentration was determined with the Pierce BCA Protein Assay Kit (Thermo Fisher Scientific).

## Mammalian cell immunofluorescence

Cells were fixed with 4% formaldehyde (Thermo Fisher Scientific) in PBS for 15 min at room-temperature. Cells were washed three times with PBS followed by blocking and permeabilization with 1% BSA (Sigma-Aldrich, Saint Louis, Missouri, USA)/0.3% Triton X-100 in PBS for 1 hr. Antibodies were diluted in blocking/permeabilization buffer and incubated for 2 hr at room temperature. Primary antibodies used were: anti-GFP (clone 5G4, Ogris lab, MFPL), anti-GRASP65 (Thermo Fisher, PA3-910), anti-Rab5 (Cell Signaling Technology, #C8B1), anti-Rab7 (Cell Signaling Technology, #D95F2)

and anti-LAMP1 (Abcam, Cambridge, UK, #ab24170). Cells were washed three times with PBS-Tween20 (0.05%) for 5 min each, followed by secondary antibody incubation in blocking/permeabilization buffer for 1 hr at room-temperature. Secondary antibodies used were: goat anti-mouse IgG (H + L) Alexa Fluor 488 (Thermo Fisher A11001), goat anti-rabbit IgG (H + L) Alexa Fluor 555 (Thermo Fisher, A21428), Cells were counterstained with DAPI (Thermo Fisher) for 10 min in PBS. Cells were mounted with ProLong Gold Antifade Mountant (Thermo Fisher #P36930). Images were acquired using a Plan-Apochromat 40x/1.4 Oil DIC objective M27 on a Zeiss LSM880 confocal microscope. Pictures were processed with ImageJ.

## Quantification of secretory pathway marker colocalization with Mrva, MFSD1 and Qsox1

Colocalization analysis was performed by ImageJ's (NIH) Coloc two plugin and determined with the pixel intensity spatial correlation analysis (Pearson's correlation coefficient).

## Embryonic protein prep for glycoproteomics

150 mg fly embryos were homogenized in 2 ml 0.1% RapiGest, 50 mM ammonium bicarbonate using a dounce homogenizer. The lysed material was left on ice for 40 min with occasional vortexing followed by probe sonication (5 s sonication, 5 s pause, 6 cycles at 60% amplitude). The lysate was cleared by centrifugation (1,000 × g for 10 min). The cleared lysate was heated at 80°C, 10 min followed by reduction with 5 mM dithiothreitol (DTT) at 60°C, 30 min and alkylation with 10 mM iodoacetamide at room temperature (RT) for 30 min before overnight (ON) digestion at 37°C with 25 μg trypsin (Roche). The tryptic digests were labeled with dimethyl stable isotopes as described (*Boersema et al., 2009*). The digests were acidified with 12 μL trifluoroacetic acid (TFA), 37°C, 20 min and cleared by centrifugation at 10,000 g, 10 min. The cleared acidified digests were loaded onto equilibrated SepPak C18 cartridges (Waters) followed by 3 × CV 0.1% TFA wash. Digests were labeled on the column by adding 5 mL 30 mM $NaBH_3CN$ and 0.2% formaldehyde ($COH_2$) in 50 mM sodium phosphate buffer pH 7.5 (Light, *mrva^3102*), or 30 mM $NaBH_3CN$ and 0.2% deuterated formaldehyde ($COD_2$) in 50 mM sodium phosphate buffer pH 7.5 (Medium, control). Columns were washed using 3 CV 0.1% FA and eluted with 0.5 mL 50% MeOH in 0.1% FA. The eluates were mixed in a 1:1 ratio, concentrated by evaporation, and resuspended in Jacalin loading buffer (175 mM Tris-HCl, pH 7.4) Glycopeptides were separated from non-glycosylated peptides by Lectin Weak Affinity Chromatography (LWAC) using a 2.8 m column packed in-house with Jacalin-conjugated agarose beads. The column was washed with 10 CVs Jacalin loading buffer (100 μL/min) before elution with Jacalin elution buffer (175 mM Tris- HCl, pH 7.4, 0.8M galactose) 4 CVs, 1 mL fractions. The glycopeptide-containing fractions were purified by in-house packed Stage tips (Empore disk-C18, 3M).

## Quantitative O-glycoproteomic strategy

The glycopeptide quantification based on M/L isotope labeled doublet ratios was evaluated to estimate a meaningful cut-off ratio for substantial changes (*Schjoldager et al., 2015*). The labeled glycopeptides produced doublets with varying ratios of the isotopic ions as well as a significant number of single precursor ions without evidence of ion pairs. Labeled samples from control *srpHemo-3xmCherry* embryos and *mrva^3102 srpHemo-3xmCherry* mutant embryos were mixed 1:1 and subjected to LWAC glycopeptide enrichment. The distribution of labeled peptides from the LWAC flow-through showed that the quantitated peptide M/L ratios were normally distributed with 99.7% falling within ±0.55 ($Log_{10}$). We selected doublets with less/more than ±0.55($Log_{10}$) value as candidates for isoform-specific O-glycosylation events.

## Mass spectrometry

EASY-nLC 1000 UHPLC (Thermo Scientific) interfaced via nanoSpray Flex ion source to an -Orbitrap Fusion mass spectrometer (Thermo Scientific) was used for the glycoproteomic study. A precursor MS1 scan (m/z 350–1,700) of intact peptides was acquired in the Orbitrap at a nominal resolution setting of 120,000. The five most abundant multiply charged precursor ions in the MS1 spectrum at a minimum MS1 signal threshold of 50,000 were triggered for sequential Orbitrap HCD-MS2 and ETD-MS2 (m/z of 100–2,000). MS2 spectra were acquired at a resolution of 50,000. Activation times were 30 and 200 ms for HCD and ETD fragmentation, respectively; isolation width was four mass

units, and one microscan was collected for each spectrum. Automatic gain control targets were 1,000,000 ions for Orbitrap MS1 and 100,000 for MS2 scans. Supplemental activation (20%) of the charge-reduced species was used in the ETD analysis to improve fragmentation. Dynamic exclusion for 60 s was used to prevent repeated analysis of the same components. Polysiloxane ions at *m/z* 445.12003 were used as a lock mass in all runs. The mass spectrometry glycoproteomics data have been deposited to the ProteomeXchange Consortium (*Vizcaíno et al., 2014*) via the PRIDE partner repository with the dataset identifier PXD011045.

## Mass spectrometry data analysis

Data processing was performed using Proteome Discoverer 1.4 software (Thermo Scientific) using Sequest HT Node as previously described (*Schjoldager et al., 2015*).

Briefly, all spectra were initially searched with full cleavage specificity, filtered according to the confidence level (medium, low and unassigned) and further searched with the semi-specific enzymatic cleavage. In all cases the precursor mass tolerance was set to six ppm and fragment ion mass tolerance to 20 mmu. Carbamidomethylation on cysteine residues was used as a fixed modification. Methionine oxidation as well as HexNAc and HexHexNAc attachment to serine, threonine and tyrosine were used as variable modifications for MS2 data. All spectra were searched against a concatenated forward/reverse *Drosophila melanogaster*-specific database (UniProt, March 2018, containing 39034 entries with 3494 canonical reviewed entries) using a target false discovery rate (FDR) of 1%. FDR was calculated using target decoy PSM validator node. The resulting list was filtered to include only peptides with glycosylation as a modification. Glycopeptide M/L ratios were determined using dimethyl 2plex method as previously described (*Schjoldager et al., 2015*)

## Statistics and repeatability

Statistical tests as well as the number of embryos/cells assessed are listed in the Figure legends. All statistical analyses were performed using GraphPad Prism and significance was determined using a 95% confidence interval. Data points from individual experiments/embryos were pooled to estimate mean and standard error of the mean. Sample size refers to biological replicates. No statistical method was used to predetermine sample size and the experiments were not randomized. For major questions, data were collected and analyzed masked. Normality was evaluated by D'Agostino and Pearson or Shapiro-Wilk normality test. Unpaired t-test or Mann-Whitney test was used to calculate the significance in differences between two groups and One-Way Anova followed by Tukey post-test or Kruskal-Wallis test followed by Conover or Dunn's post-test for multiple comparisons.

All measurements were performed in 3–38 embryos and at least 37 oocytes. Representative images shown in *Figure 1E–G,I*, *Figure 2F,I*, *Figure 3A–C*, *Figure 5A*, *Figure 6B,D and G* and *Figure 2—figure supplement 1B-J*, *Figure 3—figure supplement 1B,H,K*, *Figure 5—figure supplement 1G, K* were from separate experiments repeated 3 to 6 times. The stainings underlying *Figure 1—figure supplement 1A-M*, Figures 2Hand Figure 6-figure supplement 1C-F are from separate experiments that were repeated at least twice. Stills shown in *Figure 3I,L* and *Figure 5D* are representative images from two-photon movies, which were repeated at least 3 times.

## Acknowledgements

We thank the following for their contributions: Dr. McNew and the *Drosophila* Genomics Resource Center supported by NIH grant 2P40OD010949-10A1 for plasmids, F Mauri, J Knöblich, and L Borsig for cell lines, K Brückner, M Sixt and the Sixt lab for helpful advice, KB, P Duchek, M Poukkula, K VijayRaghavan, and the Bloomington *Drosophila* Stock Center supported by NIH grant P40OD018537 and the Vienna *Drosophila* Resource Center for fly stocks. E Ogris, S Baumgartner and J Brennecke for gifts of antibodies, L Cooley, S Munro, A Spradling and D Williams for contributing the antibodies produced by the Developmental Studies Hybridoma Bank, which was created by the Eunice Kennedy Shriver National Institute of Child Health and Human Development of the NIH, and is maintained at the University of Iowa. We thank the Life Scientific Service Units at IST Austria for technical support and assistance with microscopy and FACS analysis, and J Friml, C Guet, T Hurd and P Rangan for comments on the manuscript. KV was supported by a DOC fellowship from the Austrian Academy of Sciences. AG and AR were supported by the Austrian Science Fund (FWF) grant DASI_FWF01_P29638S, DES by Marie Curie CIG 334077/IRTIM, and AR also by Marie Curie *IIF*

GA-2012–32950 BB: DICJI. MR was supported by the NO Forschungs und BildungsgesmbH. MM and PR received funding from the European Union's Horizon 2020 research and Innovation programme under the Marie Sklodowska-Curie Grant Agreement No. 665385. SV and HC received funding from the Lundbeck Foundation, the Novo Nordisk Foundation, and the Danish Research Foundation (DNRF107). We are deeply grateful to R Lehmann in whose lab the work that served as the foundation for this project began.

## Additional information

### Funding

| Funder | Grant reference number | Author |
|---|---|---|
| Austrian Academy of Sciences | | Katarina Valoskova |
| NO Forschungs und Bildungsges.m.b.H. | | Marko Roblek |
| Horizon 2020 Framework Programme | | Michaela Misova<br>Patricia Reis-Rodrigues |
| Austrian Science Fund | DASI_FWF01_P29638S | Aparna Ratheesh |
| H2020 Marie Skłodowska-Curie Actions | IIF GA-2012-32950 BB: DICJI | Aparna Ratheesh |
| Novo Nordisk Foundation | | Sergey Y Vakhrushev<br>Henrik Clausen |
| Lundbeck Foundation | | Sergey Y Vakhrushev<br>Henrik Clausen |
| Danish National Research Foundation | DNRF107 | Sergey Y Vakhrushev<br>Henrik Clausen |
| H2020 Marie Skłodowska-Curie Actions | CIG 334077/IRTIM | Daria E Siekhaus |

The funders had no role in study design, data collection and interpretation, or the decision to submit the work for publication.

### Author contributions

Katarina Valoskova, Conceptualization, Formal analysis, Funding acquisition, Investigation, Visualization, Writing—original draft, Writing—review and editing; Julia Biebl, Conceptualization, Formal analysis, Investigation, Visualization; Marko Roblek, Conceptualization, Investigation, Writing—review and editing; Shamsi Emtenani, Formal analysis, Visualization; Attila Gyoergy, Resources, Visualization; Michaela Misova, Kateryna Shkarina, Investigation, Writing—review and editing; Aparna Ratheesh, Formal analysis, Visualization, Writing—review and editing, Funding acquisition; Patricia Reis-Rodrigues, Investigation, Visualization; Ida Signe Bohse Larsen, Methodology; Sergey Y Vakhrushev, Data curation, Formal analysis, Methodology; Henrik Clausen, Conceptualization, Supervision, Project administration, Writing—review and editing, Funding acquisition; Daria E Siekhaus, Conceptualization, Supervision, Funding acquisition, Visualization, Writing—original draft, Project administration

### Author ORCIDs

Marko Roblek http://orcid.org/0000-0001-9588-1389
Attila Gyoergy https://orcid.org/0000-0002-1819-198X
Michaela Misova http://orcid.org/0000-0003-2427-6856
Patricia Reis-Rodrigues https://orcid.org/0000-0003-1681-508X
Daria E Siekhaus http://orcid.org/0000-0001-8323-8353

### Decision letter and Author response

Decision letter https://doi.org/10.7554/eLife.41801.031
Author response https://doi.org/10.7554/eLife.41801.032

## Additional files

### Supplementary files

• Supplementary file 1. Mass spectrometric analysis of the T and Tn antigen containing O-glycoproteome from wild type and $mrva^{3102}$ mutant Stage 11–12 *Drosophila melanogaster* embryos. Each row lists an individually identified tryptically processed peptide. The 2nd–4th columns describe the analyzed peptide. The 5th, 6th, 7th and 12th are the names and accessions to Uniprot. The 8th indicates the position of the modified amino acid. The 9th indicates the number and 10th the type of glycosylation. The 11th lists the exact position and the 13th the exact description of glycosylation. The 14th is the ratio of the amount of the particular glycopeptide in the control samples (medium) over the amount in the $mrva^{3102}$ (light). The 15th is the number of missed cleavages after the tryptic digest. The 16th is the measured intensity. The 17th column shows the mass to charge ratio.

DOI: https://doi.org/10.7554/eLife.41801.025

• Supplementary file 2. All candidate proteins from the O-glycoproteome with at least 3-fold changes in T and Tn antigen in the $mrva^{3102}$ mutant. Columns list the gene name, the predicted or known function of the gene, if other T or Tn glycosites on the protein are unchanged or changed in the opposite direction, any known human ortholog (identified by BLAST), references for links to cancer and cancer invasion for the mammalian orthologs, the precise site altered, the T and Tn antigen changes observed at a particular glycosylation site, the number of glycosites on the peptide, the peptide sequence and if the glycosylation site is conserved. The site is considered conserved if the human ortholog has a serine or threonine ±5 amino acids from the *Drosophila* glycosite. References: 1 (*Göhrig et al., 2014*); 2. (*Fan et al., 2018*); 3. (*Webb et al., 1999*); 4. (C.-C. *Chiu et al., 2011*); 5. (*Huang et al., 2016*); 6. (*Matos et al., 2015*); 7. (*Cawthorn et al., 2012*); 8. (*Cao et al., 2015*) 9. (*Walls et al., 2017*); 10.(*Zhou et al., 2017*); 11. (*Linton et al., 2008*); 12. (*Bian et al., 2016*) 13. (*Zhang et al., 2016*); 14. (*Gonias et al., 2017*); 15. (*Katchman et al., 2013*; *Katchman et al., 2011*); 16. (*Stojadinovic et al., 2007*); 17. (*Zhou et al., 2016*); 18. (*Hu et al., 2018*); 19. (*Li et al., 2008*); 20. (*Senanayake et al., 2012*); 21. (*Sheu et al., 2014*); 22. (*Mao et al., 2018*); 23.(*Yokdang et al., 2016*).

DOI: https://doi.org/10.7554/eLife.41801.026

• Supplementary file 3. T or Tn antigen modified receptors from the wild-type St 11–12 *Drosophila melanogaster* embryo O-glycoproteome. Columns list the gene name for the receptor, its reported function, what kind of glycosylation we identified to be present on the receptors in the wild type sample, and what kind of glycosylation change we observed in the $mrva^{3102}$ mutant.

DOI: https://doi.org/10.7554/eLife.41801.027

• Transparent reporting form

DOI: https://doi.org/10.7554/eLife.41801.028

### Data availability

Mass spectrometry proteomics data have been deposited to the ProteomeXchange Consortium via the PRIDE partner repository with the dataset identifier PXD011045.

The following dataset was generated:

| Author(s) | Year | Dataset title | Dataset URL | Database and Identifier |
|---|---|---|---|---|
| Valoskova K, Biebl J, Larsen ISB, Vakrushev SY, Clausen H, Siekhaus DE | 2019 | T and Tn antigen glycoproteomics of early St 12 Drosophila embryos, wild type and mrva mutant | http://proteomecentral.proteomexchange.org/cgi/GetDataset?ID=PXD011045 | ProteomeXchange Consortium, PXD011045 |

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
