## [Decision Letter]

Thank you for submitting your article "A conserved MFS orchestrates a subset of O-glycosylation to facilitate macrophage dissemination and tissue invasion" for consideration by *eLife*. Your article has been reviewed by two peer reviewers, and the evaluation has been overseen by a Reviewing Editor and Marianne Bronner as the Senior Editor. The reviewers have opted to remain anonymous.

The reviewers have discussed the reviews with one another and the Reviewing Editor has drafted this decision to help you prepare a revised submission.

Summary:

In this manuscript, the Siekhaus group use the *Drosophila* system to identify a new regulator of macrophage migration and invasion. They show that macrophages in this system express high levels of T antigen, and that this expression is maintained by an atypical MFS member which they term Minerva. Loss of function studies in the fly demonstrate that Minerva is required for dissemination, and conversely they can rescue this effect by expressing the human ortholog of the gene MFSD1. In addition, they also perform O-glycoprotemics on the flies to identify the range of proteins affected in the mutant, revealing a surprisingly small number of affected proteins. Although the detailed mechanisms by which Minerva acts remain to be elucidated, the reviewers were overall enthusiastic about the quality and importance of the work. We would therefore request revisions as outlined below to further strengthen the paper.

Essential revisions:

1) Figure 2D. The in situ hybridisations of CG8602 localisation to macrophages are not convincing. These images should be improved or removed. Do the 'invasive' population of macrophages express higher levels than the non-invasive population that migrate along the vnc?

2) How specific to macrophages is this? Do other invasive migratory cell types in *Drosophila* require Minerva?

3) The authors identify Qsox1 as a target of Minerva but what is the role of Qsox1 in driving the invasion of macrophages and how does this fit with what is already known about this invasive migration?

4) Other than many co-localisation and mutant/RNAi-based data, the study rests to a significant extent on the glycoproteomic data. Here the data were analysed based on various fixed and variable modifications, including HexNAc attachment. However, in the Figure 4 and Supplementary file 1 there is the distinction between T and Tn antigen. Thus, I assume HexHexNAc was also a variable modification (subsection “Mass spectrometry Data analysis”). Was any fragment ion scan performed (e.g., for 204 or 366)? In the supplement, m/z is given as Da: however, daltons refer to mass and not mass/charge ratios – the charge status is not listed and also not the deconvoluted mass. Thus, in the absence of example figures showing, e.g., MS/MS of Qsox1 O-glycopeptides, the data are difficult to assess. Although deposited in a database, for the sake of easy reference and transparency, example glycoproteomic data should be shown in the paper.

5) Subsection “T antigen is enriched and required in invading macrophages in *Drosophila* embryos”: it is true that Fucα1,2Gal is a known ligand for UEA whether the Gal is 1,3 or 1,4-linked. However, it is an overinterpretation to say that UEA is binding a fucosylated glycan in an insect – the interaction of lectins with non-mammalian glycomes is understudied due to a lack of relevant glycan arrays. We also do not know whether UEA may bind O-Fuc on a peptide. There is no proof for 1,2-linked Fuc on any *Drosophila* glycan and, other than O-Fuc, no fucose found in an O-glycan of this species. To prove a role for fucose in UEA binding, RNAi of the GDP-fucose transporter or GDP-Fuc synthase or putative FucTs would be needed as done for the T-antigen staining with the core 1 GalT. Therefore, unless the authors have a strategy to explain the UEA binding, it is probably best to rephrase this section, perhaps deleting the sentence “Thus T antigen and a fucosylated structure are upregulated on embryonic macrophages during their invasion” and rephrasing the next sentence to "as the source of the upregulated signal in embryonic macrophages during invasion and to characterise…"

---

## [Author Response]

Essential revisions:1) Figure 2D. The in situ hybridisations of CG8602 localisation to macrophages are not convincing. These images should be improved or removed. Do the 'invasive' population of macrophages express higher levels than the non-invasive population that migrate along the vnc?

We had tried repeatedly to get better *in situs* than the ones in the paper before we submitted it, and have now tried it again with other protocols. However we have not succeeded in this. We therefore have removed our images and will just refer to the excellent BDGP in situs (subsection “An atypical MFS member acts in macrophages to increase T antigen levels”, first paragraph). By examining these or our own in situs we do not see any difference in the levels of expression of CG8602 in invading macrophages versus those on the ventral nerve cord, and have included this information. To assess this with precise quantitation would require us to be able to differentially label the two different populations, sort them, and then conduct qPCR on them. We are unable to do this as we have no specific markers or drivers for these different subsets at the moment. We can extract a few macrophages with a pipette from particular areas but not enough for such analysis.

We have observed for several of the genes that we study that although their RNA is expressed in all macrophages before invasion, their phenotype only obviously affects the few that migrate into the germband. We hypothesize that this permits flexibility as to which macrophages make their way in.

2) How specific to macrophages is this? Do other invasive migratory cell types in Drosophila require Minerva?

The main other cell type known to carry out invasive migration in *Drosophila* are the border cells, which move in the ovary from their initial location in the surrounding epithelium between the nurse cells to make their way to their final location at the edge of the oocyte (PMID 12511865). This tissue invasion is mediated by Cadherin rather than Integrin based adhesion (PMID 9971747, 24855950). Germ cells also move through a narrow region, as they emerge from the midgut through gaps in the epithelium produced by ingressing midgut cells (PMID 22619387). They are thought to move in an amoeboid, minimally adhesive migratory mode (PMID 15469969). We have examined both border cell and germ cell migration in the control and *mrva^EP3102^*mutant and find no evidence for a defect in the *mrva* mutant. We have included this data in Figure 3—figure supplement 1H-L and in the subsection “Minerva is not required for border cell invasion or germ cell migration”.

3) The authors identify Qsox1 as a target of Minerva but what is the role of Qsox1 in driving the invasion of macrophages and how does this fit with what is already known about this invasive migration?

To completely define the role Qsox1 plays in aiding macrophage invasion would require at least one more paper’s worth of work. To begin to shed some light on this question within the eight week target, we pursued three approaches: 1) Characterizing the *qsox1* mutant phenotype in greater detail through live imaging, 2) Examining where in the cell Qsox1 is localized, 3) Testing a potential effect on an ECM component. These data are shown in two new figures, Figure 5 and Figure 5—figure supplement 1, which also incorporates the data on Qsox1 that was already in the previous version of the paper in the former figure panels Figure 4I-J and Supplementary Figure 4D-F.

1) We would have loved to examine the phenotype of Qsox1 mutated (S/T to A) at the amino acids whose T antigen modification is dependent on Minerva. However these flies are still under construction. We therefore conducted 2-photon imaging of the *qsox1* P element mutant and examined the effect on invasion (Figure 5—video 5, stills in Figure 5D). We observed that Qsox1 is required for entry into the germband (Figure 5F), with macrophages in the *qsox1* mutant pausing twice as long (49 minutes in the mutant vs. 22 minutes in the control) before entering. We have also added a figure panel showing this data for the *mrva* mutant (Figure 3K, in which entry takes on average 102 minutes). We also see a decrease in the speed of initial dissemination in the head (Figure 5E) in the *qsox1* mutant with again a similar but not as strong effect as in the *mrva* mutant (Figure 3J).

2) We find that Qsox1 when expressed in S2R+ cells localizes to the Golgi, endosomes and can be secreted (Figure 5H-I, Figure 5—figure supplement 1G-J), indicating a potential for extracellular functions. We also observe colocalization with Mrva.

3) Guided by the literature which has shown a role for Qsox1 in vertebrates in the extracellular deposition and anchoring of Laminin into the ECM (PMID 23704371), we examined the localization of Laminin A (LanA) and observed higher levels inside macrophages in *mrva* and *qsox1* mutant embryos and somewhat higher levels adjacent but not directly next to the macrophage cell edge (Figure 5J, Figure 5—figure supplement 1K-N). These data are consistent with a role for Qsox1 in speeding the secretion of LanA, and potentially allowing its deposition and anchoring in the ECM.

We describe these results in the subsection “Minerva raises T antigen levels on proteins required for invasion”, and discuss them in the subsection “An invasion program regulated by Minerva” in the context of a model integrating other data in which Qsox1 dependent-integration of normal levels of LanA into the ECM is required to allow macrophages to anchor themselves and exert traction forces that speed initial tissue invasion.

4) Other than many co-localisation and mutant/RNAi-based data, the study rests to a significant extent on the glycoproteomic data. Here the data were analysed based on various fixed and variable modifications, including HexNAc attachment. However, in the Figure 4 and Supplementary file 1 there is the distinction between T and Tn antigen. Thus, I assume HexHexNAc was also a variable modification (subsection “Mass spectrometry Data analysis”).

You are correct, and we have amended the Materials and methods to state this. Thank you for noticing this. “Methionine oxidation as well as HexNAc and HexHexNAc attachment to serine, threonine and tyrosine were used as variable modifications for MS2 data.”

Was any fragment ion scan performed (e.g., for 204 or 366)?

No additional fragment ion scans were performed. As is written in the Materials and methods section (see below), all precursors were fragmented with both ETD and HCD fragmentation:

“The five most abundant multiply charged precursor ions in the MS1 spectrum at a minimum MS1 signal threshold of 50,000 were triggered for sequential Orbitrap HCD-MS2 and ETD-MS2 (m/z of 100–2,000)."

In the supplement, m/z is given as Da: however, daltons refer to mass and not mass/charge ratios – the charge status is not listed and also not the deconvoluted mass.

We thank the reviewers for catching this error, which arose due to the default assignment by the Proteome Discoverer software (Thermo). We have now replaced Da with "Th" on column Q in Supplementary file 1, the column which corresponds to the precursor ions’ m/z. A few additional parameters such as charge, charge deconvoluted mass (M+H^+^), δ m/z (ppm) and Xcorr were added as well.

Thus, in the absence of example figures showing, e.g., MS/MS of Qsox1 O-glycopeptides, the data are difficult to assess. Although deposited in a database, for the sake of easy reference and transparency, example glycoproteomic data should be shown in the paper.

We have now added the MS/MS traces for the Qsox1 O-glycopeptides to the paper in Figure 4I and Figure 4—figure supplement 1B-C.

5) Subsection “T antigen is enriched and required in invading macrophages in Drosophila embryos”: it is true that Fucα1,2Gal is a known ligand for UEA whether the Gal is 1,3 or 1,4-linked. However, it is an overinterpretation to say that UEA is binding a fucosylated glycan in an insect – the interaction of lectins with non-mammalian glycomes is understudied due to a lack of relevant glycan arrays. We also do not know whether UEA may bind O-Fuc on a peptide. There is no proof for 1,2-linked Fuc on any Drosophila glycan and, other than O-Fuc, no fucose found in an O-glycan of this species. To prove a role for fucose in UEA binding, RNAi of the GDP-fucose transporter or GDP-Fuc synthase or putative FucTs would be needed as done for the T-antigen staining with the core 1 GalT. Therefore, unless the authors have a strategy to explain the UEA binding, it is probably best to rephrase this section, perhaps deleting the sentence “Thus T antigen and a fucosylated structure are upregulated on embryonic macrophages during their invasion” and rephrasing the next sentence to "as the source of the upregulated signal in embryonic macrophages during invasion and to characterise…"

We expressed the VDRC RNAi line 30238 against the fucose transporter Efr but did not see any reduction in UEA binding. We did not have time to grow a large enough stock of flies to be able to conduct qPCR analysis on the FACS sorted macrophages from the RNAi expressing embryos to determine if the knockdown was effective and thus to see if we could conclude that UEA does not recognize fucosylated glycan structures in the insect. We have therefore changed the text in the paragraph as suggested by the reviewers.